# Relative Entropic Optimal Transport: a (Prior-aware) Matching Perspective to (Unbalanced) Classification

**Liangliang Shi, Haoyu Zhen, Gu Zhang, Junchi Yan**[*]
Dept. of Computer Science and Engineering & MoE Key Lab of AI, Shanghai Jiao Tong University
{shiliangliang, anye_zhen, blake-nash, yanjunchi}@sjtu.edu.cn
PyTorch Code: https://github.com/LiangliangShi/RE-OT

## Abstract

Classification is a fundamental problem in machine learning, and considerable efforts have been recently devoted to the demanding long-tailed setting due to its prevalence in nature. Departure from the Bayesian framework, this paper rethinks classification from a matching perspective by studying the matching probability between samples and labels with optimal transport (OT) formulation. Specifically, we first propose a new variant of optimal transport, called Relative Entropic Optimal Transport (RE-OT), which guides the coupling solution to a known prior information matrix. We gives some theoretical results and their proof for RE-OT and surprisingly find RE-OT can help to deblur for barycenter images. Then we adopt inverse RE-OT for training long-tailed data and find that the loss derived from RE-OT has a similar form to Softmax-based cross-entropy loss, indicating a close connection between optimal transport and classification and the potential for transferring concepts between these two academic fields, such as barycentric projection in OT, which can map the labels back to the feature space. We further derive an epoch-varying RE-OT loss, and do the experiments on unbalanced image classification, molecule classification, instance segmentation and representation learning. Experimental results show its effectiveness.

## 1 Introduction

Long-tailed label distribution or unbalanced classification in training given more balanced testing set has been a challenge for existing neural models [42, 3, 62]. Various approaches have been developed, such as Class Re-balancing [3, 25, 35, 8, 45, 52], Information Augmentation [36, 7], and Module Improvement [65, 16, 62]. However, the problem is still far from being solved.

Most existing methods on (long-tailed) classification study the conditional probability function $p(y|x)$ or their joint distribution given the label $y$ and sample $x$, almost taking it for granted. For instance, [45] shows that in the long-tailed case, according to Bayes' theorem, the vanilla Softmax classification is affected by the label distribution shift, which makes the classifier more inclined to consider the samples as belonging to the majority class. To address this issue, the work explicitly considers the label distribution shift and re-derives the Softmax function. [22] argues that the conditional distribution $p(y|x)$ is not the same between training and testing data, which can result in target-shift. To enhance class-unbalanced learning, the authors propose to relax the assumption that the source and target domains share the same conditional distribution $p(x|y)$. Despite the de-facto Bayesian perspective for understanding classification, in this paper, we take a matching view and demonstrate how new insights and approaches can be derived, especially for the challenging long-tail cases.

---

[*]Correspondence author. This work was partly supported by NSFC (62222607, 61972250, U19B2035) and Shanghai Municipal Science and Technology Major Project (2021SHZDZX0102).

From the matching perspective, the problem becomes matching samples to labels. During training, the goal is to learn the distances between the features of samples and labels. During testing, matching results can be obtained by calculating the distances between the features of testing samples and their labels. We formulate this matching using the theory of Optimal Transport (OT), where training precisely involves learning the Inverse OT [12, 33, 51], and testing involves optimizing the transportation results. In the commonly studied long-tailed setting [35], the labels of training samples follow a long-tail distribution, while the labels are uniformly distributed for testing data. With our OT formulation, we demonstrate that these two different priors (i.e. long-tailed and uniform) can effectively reduce the difference between training and testing, thereby increasing the accuracy of the matching probability.

To incorporate the prior information e.g. the aforementioned typical long-tailed setting, we introduce the smoothing-guided prior matrix $\mathbf{Q}$ to vanilla entropic OT and call the resulting formulation relative entropic OT (RE-OT). To this new variant of OT, we provide convergence analysis, its corresponding static Schrödinger form, Sinkhorn algorithm for solving the problem as well as the dual formulation of RE-OT, which are mostly similar to those for the vanilla entropic OT. We find our RE-OT can help to deblur for barycenter images by setting specific smoothing prior $\mathbf{Q}$.

Specifically, we develop the inverse RE-OT (as our relative entropy OT's counterpart to inverse OT) to learn the cost matrix or equivalently, the sample features for training data. And then we use the RE-OT to optimize the transportation to get the matching results (i.e. predictions for testing data). It is uncovered that many Softmax-based cross-entropy losses (e.g. Balanced Softmax, A-softmax and etc) are special cases of our derived new loss under half constraints of OT by specifying different forms of the cost matrix, supervision matrix in IOT and the smoothing-guided prior matrix $\mathbf{Q}$. In other words, we can generalize the cross-entropy loss by specifying different cost matrix, smoothing-guided matrix, or other settings in the framework of inverse OT. We also find that when the regularization coefficient $\epsilon \rightarrow 0$, the loss of inverse RE-OT equals triplet loss in metric learning fields. Additionally, our matching perspective reveals interesting connections between OT and classification tasks. For example, we discover the equivalence between the coefficient of entropic regularization in OT and the temperature in Softmax. We also find that the concept of barycentric projection in OT can be used in classification to calculate the class barycenters in hidden feature space. We believe that the connection between these two fields will inspire further algorithmic development for both sides. **This paper contributes in the following ways:**

1) We propose a matching perspective to revisit and provide new understanding to the classification problem with OT formulation. The matching perspective provide a clear interpretation to the challenge of long-tailed classification, which lies in the inconsistency of the utilized prior information between the matches learned from the training set and those from the testing set.

2) We propose a new variant of entropic optimal transport, called relative entropic OT (RE-OT), for learning matching with a specified prior. We provide theoretical results on various aspects of RE-OT, including its solution properties, Sinkhorn algorithm, barycenter calculation, and its dual form. As a side product of our study, we find that RE-OT can help to deblur barycenter images by setting a specific smoothing prior $\mathbf{Q}$, which entropic OT alone cannot achieve.

3) We further develop Inverse RE-OT to link our results to classification, and show that Softmax and its variants are special cases of our derived loss based on Inverse RE-OT. This finding inspires a simple yet effective technique in practice for long-tailed classification: we let the smoothing-guided matrix $\mathbf{Q}$ to be epoch-varying, which gradually provides prior information and improves the performance.

4) Our study has revealed a significant link between Optimal Transport (OT) and existing classification tasks, such as the relationship between the regularization coefficient in OT and the temperature in Softmax. This suggests that OT concepts can be applied to classification tasks, and vice versa. For instance, the barycentric projection concept in OT can be utilized in classification tasks to calculate the class barycenters in the hidden feature space. We anticipate that this connection between the two fields will lead to further algorithmic development for both OT and classification.

5) Our method performs competitively and often outperforms baselines on various long-tailed benchmarks ranging from image classification, to molecule property prediction, and instance segmentation. Source code will be made publicly available.

## 2 Preliminaries and Related Work

**Optimal Transport.** Known as Wasserstein distance or Earth Mover's distance, Optimal transport (OT) [26], has gained significant attention in various fields including domain adaption [11], generative models [15, 34, 49], and image registration [14]. We begin by briefly introducing discrete Optimal Transport (OT), which provides a framework for viewing the (long-tailed) classification problem from a matching perspective. For a more detailed introduction to OT, readers are directed to [43].

Consider two histograms $\mathbf{a} \in \Sigma_n$ and $\mathbf{b} \in \Sigma_m$, where the simplex $\Sigma_d = \{x \in \mathbb{R}_+^d | x^\top 1_d = 1\}$. We can represent the transportation polytope $U(\mathbf{a}, \mathbf{b})$, which is the polyhedral set of $n \times m$ matrices:

$$U(\mathbf{a}, \mathbf{b}) = \{\mathbf{P} \in \mathbb{R}_+^{n \times m} | \mathbf{P1}_m = \mathbf{a}, \mathbf{P}^\top 1_n = \mathbf{b}\}, \tag{1}$$

where $\mathbf{1}_n$ and $\mathbf{1}_m$ are $n$ and $m$ dimensional vectors of ones. $U(\mathbf{a}, \mathbf{b})$ contains all $n \times m$ nonnegative probabilistic matrices with row and column sums $\mathbf{a}$ and $\mathbf{b}$. Then the Kantorovich's optimal transport problem can be defined as [26]:

$$\min_{\mathbf{P} \in U(\mathbf{a}, \mathbf{b})} < \mathbf{C}, \mathbf{P} >= \sum_{i,j} \mathbf{C}_{ij} \mathbf{P}_{ij}, \tag{2}$$

where $\mathbf{C} \in \mathbb{R}_+^{n \times m}$ is the cost matrix for the distance between sample $\{\mathbf{x}_i\}_{i=1}^n$ and $\{\mathbf{y}_j\}_{j=1}^m$. The above minimization is a linear program and the optimal solution $\mathbf{P}^*$ can be obtained with the network simplex [54] or other off-the-shelf techniques e.g. [40], which require significant time overhead. For its cost-effectiveness, the entropic regularization [57] is more popular and it gets an approximation:

$$\min_{\mathbf{P} \in U(\mathbf{a}, \mathbf{b})} \mathcal{L}^\epsilon =< \mathbf{C}, \mathbf{P} > -\epsilon H(\mathbf{P}), \tag{3}$$

where the regularizer is specified as $H(\mathbf{P}) = \sum_{ij} -\mathbf{P}_{ij}(\log(\mathbf{P}_{ij}) - 1)$. It can be proved that [43] when $\epsilon \to 0$, the unique solution of Eq. 3 converges to the optimal $\mathbf{P}^*$ of original problem. And when $\epsilon \to +\infty$, we can get

$$\mathbf{P}_\epsilon \overset{\epsilon \to +\infty}{\longrightarrow} \mathbf{a} \otimes \mathbf{b}, \tag{4}$$

where $\mathbf{a} \otimes \mathbf{b} = \mathbf{a}^\top \mathbf{b} \in U(\mathbf{a}, \mathbf{b})$. So we can find the entropic regularization $H(\mathbf{P})$ exactly do a smoothing to the solution towards $\mathbf{a} \otimes \mathbf{b}$. Recently, Inverse Optimal Transport (IOT) start to be studied [12, 33, 51, 50] which assumes that the cost $\mathbf{C}$ is unknown for learning, given the established coupling. The inverse of entropic OT problem can be formulated as:

$$\min_\theta KL(\tilde{\mathbf{P}}|\mathbf{P}^\theta), \quad \text{where} \quad \mathbf{P}^\theta = \arg \min_{\mathbf{P} \in U(\mathbf{a}, \mathbf{b})} < \mathbf{C}^\theta, \mathbf{P} > -\epsilon H(\mathbf{P}). \tag{5}$$

Specifically, [51] proposes a principled approach to infer the unknown costs and [6] develops the mathematical theory for IOT. [33] shows that by IOT, one can also modify and improve the raw matching prediction by the cost, which is learned by the raw matching as noise supervision.

In this paper, we develop an IOT-based formulation to learn the cost between samples and their labels, which further allows one to learn the feature extractor of the training samples. Then with the features exacted from testing data, we match them with target labels and achieve the goal of classification.

**Long-tailed recognition.** Long-tailed recognition has been a standing problem with wide attention in vision and learning [64, 19, 35]. One common approach [45, 41] is to re-balance the class distribution in the training set. For example, some works employ re-sampling [25] or re-weighting [8] techniques to let the model pay more attention to the minority classes in the training set. Data augmentation is also adopted to synthesize more samples for the underrepresented classes [28]. While in [9], transfer learning is used, which leverages knowledge learned from a source domain with sufficient data to improve the performance on a target domain with imbalanced data. This line of research includes techniques such as fine-tuning [63], domain adaptation [53], and meta-learning [56]. Recently, some works have proposed to explicitly model the long-tailed distribution, such as by adjusting the loss or designing specialized network architectures. Examples include focal loss [35], class-balanced loss [8], balanced Softmax loss [45]. In this paper, we view classification from a matching perspective. Our method does not predict the label or class directly but learns the features in training and calculates the matching probability instead. With the aid of OT, we formulate it with relative entropic as proposed in the next section, which adopts different prior for training and testing.

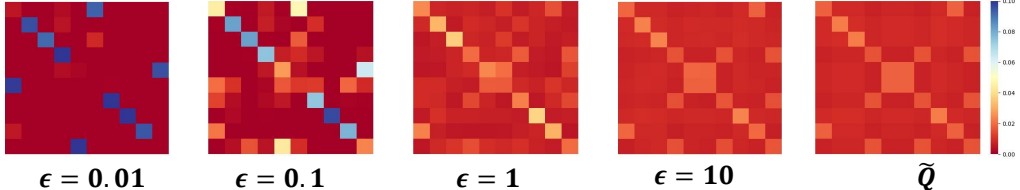

$\epsilon = 0.01$  $\quad$  $\epsilon = 0.1$  $\quad$  $\epsilon = 1$  $\quad$  $\epsilon = 10$  $\quad$  $\widetilde{Q}$

Figure 1: Results of $\mathbf{P}_{\mathbf{Q}}^{\epsilon}$ by varying $\epsilon$ given the features (trained with SimCLR [5]) of ten CIFAR-10 images and their augmentation. The cost matrix is set with the cosine distance between features. When $\epsilon \to 0$, $\mathbf{P}_{\mathbf{Q}}^{\epsilon}$ becomes sharper for probability prediction and when $\epsilon \to +\infty$, $\mathbf{P}_{\mathbf{Q}}^{\epsilon}$ is more close to $\widetilde{\mathbf{Q}}$. Here we set $\mathbf{Q}_{ij} = 0.7$ if sample $i$ and $j$ are in the same class and $\mathbf{Q}_{ij} = 0.3$ otherwise.

## 3   Relative Entropy Regularization for Optimal Transport

**Motivation of introducing prior into OT.** As introduced in Eq. 3, entropic regularization (ER) is an effective method [10] for approximating the Kantorovich OT solution. It pushes the original linear programming solution away from the hard boundary and obtains a smoother solution by minimizing a $\epsilon-$strongly convex function. With the increase of the penalty coefficient, the solution progressively approaches a more 'smooth' solution, ultimately converging to $\mathbf{a}^\top \mathbf{b}$ as illustrated by Eq. 4. However, a question is raised: is such a smoothing direction $\mathbf{a}^\top \mathbf{b}$ appropriate for all practical situations, e.g. the long-tailed problems? Can we freely choose a more suitable (smoothing) direction to achieve more effective results tailored to the problem or dataset at hand? In this section, we will develop relative entropic OT (RE-OT), a more general formulation than traditional entropic OT [57].

**Formulation.** Given a cost matrix $\mathbf{C} \in \mathbb{R}_+^{n \times m}$ and two margins $\mathbf{a} \in \Sigma_n$ and $\mathbf{b} \in \Sigma_m$, where $\Sigma_n$ and $\Sigma_m$ represent the probability simplex with $n$ and $m$ bins, respectively, we introduce the positive smoothing-guidance matrix $\mathbf{Q}$ as a manually-specified constant prior to guide the smoothing. We use the relative entropy regularizer $H_{\mathbf{Q}}(\mathbf{P})$ in place of the traditional entropy in Eq. 3 to achieve this ability, defined as $H_{\mathbf{Q}}(\mathbf{P}) = -\sum_{ij} \mathbf{P}_{ij} \left( \log \frac{\mathbf{P}_{ij}}{\mathbf{Q}_{ij}} - 1 \right)$. Then we have:

$$\min_{\mathbf{P} \in U(\mathbf{a}, \mathbf{b})} \mathcal{L}_{\mathbf{Q}}^{\epsilon} = <\mathbf{C}, \mathbf{P}> -\epsilon H_{\mathbf{Q}}(\mathbf{P}). \tag{6}$$

The relative entropy $H_{\mathbf{Q}}(\mathbf{P})$ can increase the probability of transportation for the element $\mathbf{P}_{ij}$ if $\mathbf{Q}_{ij}$ is larger. Consequently, if the prior in the form of $\mathbf{Q}$ is available, the use of relative entropy regularization can improve the efficiency of learning the transportation solution. Therefore, for long-tailed tasks, we can assign a large $\mathbf{Q}$ element to the majority classes and a low element of $\mathbf{Q}$ to the tailed classes to enhance training. We will discuss this in detail in the next section.

For Eq. 6, we further study its properties. Firstly, we show the relation between RE-OT and the entropic OT by Eq. 3 as follows. The proof is given in Table A.

**Proposition 1** (**Solution Propriety.**). *Assume* $\mathbf{P}_Q^\epsilon, \mathbf{P}^\epsilon$ *and are the optimal solution of RE-OT and entropic regularized OT, respectively. Then we can get:*

*(1) When* $\epsilon \to +\infty$*, the optimal RE-OT's solution* $\mathbf{P}_Q^\epsilon$ *will converge to* $\widetilde{\mathbf{Q}}$ *where* $\widetilde{\mathbf{Q}}$ *takes the form* $\widetilde{\mathbf{Q}} = diag(\mathbf{u})\mathbf{Q}diag(\mathbf{v})$ *with two uniquely defined non-negative vectors* $\mathbf{u}$ *and* $\mathbf{v}$*.*

*(2) With the prior* $\mathbf{Q}$ *and its corresponding* $\widetilde{\mathbf{Q}}$ *as defined in (1), we have* $\mathbf{P}_{\mathbf{Q}}^\epsilon = \mathbf{P}_{\widetilde{\mathbf{Q}}}^\epsilon$*. And when* $\widetilde{\mathbf{Q}} = \mathbf{a} \otimes \mathbf{b}$*, we have the equality* $\mathbf{P}^\epsilon = \mathbf{P}_{\widetilde{\mathbf{Q}}}^\epsilon$*.*

Proposition 1 shows the relation between RE-OT and entropic OT, and we can find RE-OT is a more general formulation. Note $\widetilde{\mathbf{Q}}$ is defined by $\mathbf{Q}$ and can be obtained by iteratively normalizing row/column sums of $\mathbf{Q}$. Fig. 1 shows the results of $\mathbf{P}_{\mathbf{Q}}^\epsilon$ by varying $\epsilon$. As $\epsilon$ increases, the optimal coupling becomes more dense, but $\mathbf{P}^\epsilon$ becomes more uniform while $\mathbf{P}_Q^\epsilon$ converges to $\widetilde{\mathbf{Q}}$.

Besides, similar to the standard entropic regularized OT, our RE-OT in Eq. 6 can also be reformulated to the "static Schrödinger problem" form [31], which exactly leans a projection under KL divergence.

**Proposition 2** (**static Schrödinger form**). *Redefine a general KL divergence as*

$$\widetilde{KL}(\mathbf{P}|\mathbf{K}) = \sum_{ij} \mathbf{P}_{ij} \log \frac{\mathbf{P}_{ij}}{\mathbf{K}_{ij}} - \mathbf{P}_{ij} + \mathbf{K}_{ij}, \tag{7}$$

*The optimization in Eq. 6 is equivalent to the following minimization, where $\mathbf{K}_{ij} = \mathbf{Q}_{ij} e^{-\mathbf{C}_{ij}/\epsilon}$ :*

$$\mathbf{P}_Q^\epsilon = \underset{\mathbf{P} \in U(\mathbf{a},\mathbf{b})}{\arg\min} \widetilde{KL}(\mathbf{P}|\mathbf{K}). \tag{8}$$

The proof is in Appendix B. Prop. 2 shows the optimal solution $\mathbf{P}_Q^\epsilon$ is exactly the KL projection of $\mathbf{K}$ onto $U(\mathbf{a}, \mathbf{b})$. In fact, $\mathbf{K}$ can be set as $\mathbf{K}_{ij} = \widetilde{\mathbf{Q}}_{ij} e^{-\mathbf{C}_{ij}/\epsilon}$ due to the equivalence between $\mathbf{P}_{\mathbf{Q}}^\epsilon$ and $\mathbf{P}_{\widetilde{\mathbf{Q}}}^\epsilon$. Expect for the different form of $\mathbf{K}$, RE-OT and entropic OT share the same static Schrödinger formulation, which means that many methods in entropic OT can be applied to RE-OT, such as iterative Bregman projections [2].

**The Sinkhorn algorithm and barycenters for RE-OT.** The optimal solution for RE-OT can be estimated using the Sinkhorn algorithm, as its counterpart has already been well-developed for solving entropic OT [10]. The algorithm for RE-OT shares the same form as that for entropic OT, given $\mathbf{K}_{ij} = \widetilde{\mathbf{Q}}_{ij} e^{-\mathbf{C}_{ij}/\epsilon}$. Specifically, with two non-negative vectors $\mathbf{u}'$ and $\mathbf{v}'$ uniquely defined up to a multiplicative factor, the optimal solution has the form $\mathbf{P}_Q^\epsilon = \text{diag}(\mathbf{u}') \mathbf{K} \, \text{diag}(\mathbf{v}')$, which can be efficiently computed by iterating $\mathbf{u}', \mathbf{v}' \leftarrow \mathbf{a}./\mathbf{K}\mathbf{v}', \mathbf{b}./\mathbf{K}^\top \mathbf{u}'$. The proof and algorithm are presented in Appendix C. In addition, the barycenter between distributions is a natural extension of OT [1, 2], and with the matrix $\mathbf{K}$, [2] defined it with a weighted KL projection problem:

$$\min_{\{\mathbf{P}_s\},\mathbf{a}} \sum_s \epsilon \cdot \lambda_s \widetilde{KL}(\mathbf{P}_s|\mathbf{K}) \text{ s.t. } \mathbf{P}_s^\top \mathbf{1}_n = \mathbf{b}_s, \mathbf{P}_1 \mathbf{1}_m = \mathbf{P}_2 \mathbf{1}_m = \cdots = \mathbf{P}_S \mathbf{1}_m = \mathbf{a}, \tag{9}$$

where $\{\lambda_s\}_{s=1}^S$ are the weights, $\mathbf{b}_s$ are known histograms representing images, and $\mathbf{a}$ is the barycenter histogram to be calculated. Though easier to calculate compared with the regularized formulation in [1], a drawback of this entropic-based method is that it can lead to blurred barycenters, and methods have been proposed to address this issue [23]. By calculating barycenters between noise and an image (i.e. $S = 2$ here), we show that the blurred problem can be simply solved by setting $\lambda$ where $\lambda_1 = \lambda, \lambda_2 = 1 - \lambda$ and the prior matrix $\mathbf{Q}$ :

$$\mathbf{Q} = (1 - \lambda)(\mathbf{P}^\epsilon)^\top + \lambda \mathbf{P}^\epsilon, \tag{10}$$

where $\mathbf{P}^\epsilon$ is the optimal solution of entropic OT from image $\mathbf{b}_1$ to $\mathbf{b}_2$ . Fig. 2 illustrates the effect of different choices of $\mathbf{Q}$ when computing barycenters between noise and an image. Without using $\mathbf{Q}$, the resulting barycenters are very blurry. When we set $\mathbf{Q}$ to be $\mathbf{P}^\epsilon$, the barycenters display the shape of the leopard clearly, but fail to transition smoothly from noise to leopard image as $\lambda$ changes. To address this issue, we set $\mathbf{Q}$ as shown in Eq. 10, which can blur the image while transitioning smoothly with the gradient.

**Proposition 3** (**Dual formulation**). *From the optimization in Eq. 6, we can get its dual formulation:*

$$L_{\mathbf{Q}}^\epsilon = <\mathbf{f}, \mathbf{a}> + <\mathbf{g}, \mathbf{b}> -\epsilon < e^{\mathbf{f}/\epsilon}, \mathbf{K} e^{\mathbf{g}/\epsilon} > \tag{11}$$

*where $\mathbf{f} \in \mathbb{R}^n$ and $\mathbf{g} \in \mathbb{R}^m$ are the corresponding dual variables.*

The proof is given in Appendix D. Exactly $\mathbf{f}, \mathbf{g}$ are linked to $\mathbf{u}, \mathbf{v}$ appearing in Sinkhorn algorithm by $\mathbf{u} = \exp(\mathbf{f}/\epsilon)$ and $\mathbf{v} = \exp(\mathbf{g}/\epsilon)$, and thus Sinkhorn algorithm can be done in log-domain [47].

**Setting Q with the Optimal Solution Iteratively.** The intuition for the prior is to iteratively update the solution $P_{\mathbf{Q}}^\epsilon$ as a new $\mathbf{Q}$, i.e., $\mathbf{Q}^{(t)} = \mathbf{P}_{\mathbf{Q}^{(t-1)}}^\epsilon$, where $\mathbf{Q}^{(t)}$ represents the solution obtained after the $n$-th iteration. The question then arises as to how $\mathbf{Q}^{(t)}$ changes over time. We find that this problem is equivalent to a proximal point algorithm:

$$\mathbf{Q}^{(t)} = \underset{\mathbf{Q}}{\arg\min} \, KL(\mathbf{Q}|\mathbf{Q}^{(t-1)}) + \frac{1}{\epsilon} F(\mathbf{Q}). \tag{12}$$

Here $F(\mathbf{Q}) = <C, \mathbf{Q}> + l_{U(\mathbf{a},\mathbf{b})}(\mathbf{Q})$, where $l_{U(\mathbf{a},\mathbf{b})}(\mathbf{Q})$ is defined such that $l_{U(\mathbf{a},\mathbf{b})}(\mathbf{Q}) = 0$ if $\mathbf{Q} \in U(\mathbf{a}, \mathbf{b})$, and $l_{U(\mathbf{a},\mathbf{b})}(\mathbf{Q}) = +\infty$ otherwise. We find this problem is discussed by [29, 43],

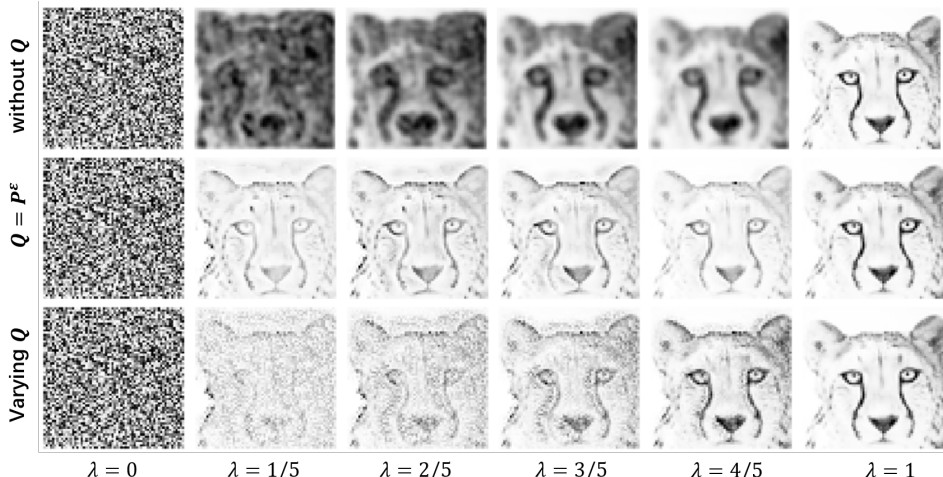

On the left side, vertical labels read: **without $Q$**, **$Q = P^\epsilon$**, **Varying $Q$**

Bottom labels: $\lambda = 0$    $\lambda = 1/5$    $\lambda = 2/5$    $\lambda = 3/5$    $\lambda = 4/5$    $\lambda = 1$

Figure 2: The barycenter results between noise and a leopard image. It is very blurry without the use of $\mathbf{Q}$. However, when we set $\mathbf{Q} = \mathbf{P}^\epsilon$, the resulting barycenters clearly display the shape of the leopard, but fail to transition smoothly from noise to leopard image as $\lambda$ changes. To address this, we set $\mathbf{Q}$ as shown in Eq. 10, which blurs the image while transitioning with the increase of $\lambda$.

which aims to get the optimal solution of the non-regularized OT. As discussed in Sinkhorn of RE-OT, we have the optimal solution form:

$$
\begin{aligned}
\mathbf{Q}^{(t)} &= \operatorname{diag}(\mathbf{u}^{(t-1)})\mathbf{Q}^{(t-1)} \odot e^{-C/\epsilon} \operatorname{diag}(\mathbf{v}^{(t-1)}) \\
&= \operatorname{diag}(\mathbf{u}^{(t-1)} \odot ... \odot \mathbf{u}^{(0)})\mathbf{Q}^{(0)} \odot e^{-\frac{(t+1)C}{\epsilon}} \operatorname{diag}(\mathbf{v}^{(0)} \odot ...\mathbf{v}^{(t-1)}).
\end{aligned}
\tag{13}
$$

When we set $\mathbf{Q}^{(0)} = \mathbf{1}_{n \times m}$, the optimization is equivalent to applying Sinkhorn's algorithm iteratively with a kernel $e^{-\frac{(t+1)C}{\epsilon}}$, i.e., with a decaying regularization coefficient $\frac{\epsilon}{t+1}$. This observation further emphasizes the importance of selecting an appropriate $\mathbf{Q}$ for RE-OT.

## 4  RE-OT for Long-tailed Recognition

We now discuss the application of inverse RE-OT to address the long-tailed classification task (including the contrastive learning setting). Unlike previous works that study this problem using conditional probability, our approach aims to learn features by matching samples with their labels using Inverse OT. Specifically, given the mini-batch pair set $\{(\mathbf{x}_i, \mathbf{y}_i)\}_{i=1}^{n}$ where $\mathbf{x}_i$ is training sample and $\mathbf{y}_i$ is the $m-$dimensional one-hot label of $\mathbf{x}_i$. We can set the the cost $\mathbf{C}_{ij}^\theta$ between sample $\mathbf{x}_i$ and another one-hot label given the neural network $f_\theta$. Without loss of generality, we can set $\mathbf{C}_{ij}^\theta = c - l_{ij}$ where $l_{ij} = (f_\theta(\mathbf{x}_i))_j$ is the $j-$th component of logits

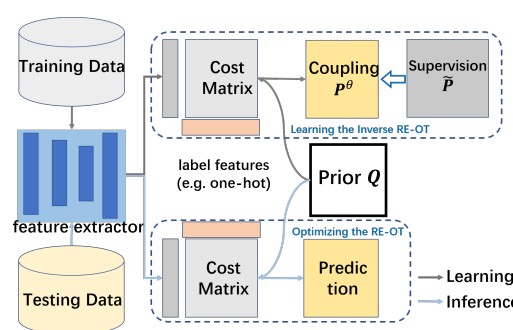

Figure 3: RE-OT for unbalanced classification.

$f_\theta(\mathbf{x}_i)$ for sample $\mathbf{x}_i$. Then we can learn the matching between features and labels with inverse OT via a bi-level optimization:

$$
\min_\theta KL(\widetilde{\mathbf{P}} | \mathbf{P}^\theta) \quad s.t. \quad \mathbf{P}^\theta = \operatorname*{arg\,min}_{\mathbf{P} \in U} <\mathbf{C}^\theta, \mathbf{P}> -\epsilon H_\mathbf{Q}(\mathbf{P}),
\tag{14}
$$

where $\widetilde{\mathbf{P}}$ is the supervision given the label $\{\mathbf{y}_i\}$. For example, we can set $\widetilde{\mathbf{P}}_{ij} = \mathbf{y}_{ij}$ when $\mathbf{y}_{ij}$ is the $j$-th component of $\mathbf{y}_i$. Here $U$ refer to the constraints for the coupling $\mathbf{P}$. We can set $U = U(\mathbf{a}, \mathbf{b})$ with the full matching constraints or its relaxation e.g. $U = U(\mathbf{a}) = \{\mathbf{P} | \mathbf{P} \mathbf{1}_m = \mathbf{a}\}$.

For our designed smoothing guidance $\mathbf{Q}$, a simple idea is to set it with the long-tailed ratio i.e. the training sample ratio of class labels which are readily available from the training set, which equals to

Table 1: Previous works in the view of inverse RE-OT by setting different ground truth $\tilde{\mathbf{P}}_{ij}$, cost $\mathbf{C}_{ij}$, prior $\mathbf{Q}_{ij}$ and coefficient $\epsilon$ under the constraints $U(\mathbf{a})$.

| Methods | Formulation | | | |
|---|---|---|---|---|
| | Ground Truth $\tilde{\mathbf{P}}_{ij}$ | Cost $\mathbf{C}_{ij}$ | Prior $\mathbf{Q}_{ij}$ | penalty coefficient $\epsilon$ |
| Vanilla Softmax | $\tilde{\mathbf{P}}_{ij} = y_{ij}$ | $\mathbf{C}_{ij} = c - l_{ij}$ | $Q_{ij} = 1$ | $\epsilon = \tau$ |
| Focal Loss [35] | $\tilde{\mathbf{P}}_{ij} = y_{ij} * (1 - \mathbf{P}_{ij})^{\gamma}$ | $\mathbf{C}_{ij} = c - l_{ij}$ | $Q_{ij} = 1$ | $\epsilon = \tau$ |
| A-softmax [37] | $\tilde{\mathbf{P}}_{ij} = y_{ij}$ | $\mathbf{C}_{ij} = c - \|l_i\|\psi(\theta_{\mathbf{y},i})$ | $Q_{ij} = 1$ | $\epsilon = \tau$ |
| Triplet [48] | $\tilde{\mathbf{P}}_{ij} = y_{ij}$ | $\mathbf{C}_{ij} = \|f(x_i) - f(x_j)\|^2$ | $Q_{ij} = 1$ | $\epsilon \to 0^+$ |
| Class-Balanced Loss [8] | $\tilde{\mathbf{P}}_{ij} = (1-\beta)/\left(1 - \beta_{y_{i,:}}^n\right)$ | $\mathbf{C}_{ij} = c - l_{ij}$ | $Q_{ij} = 1$ | $\epsilon = \tau$ |
| LDAM Loss [3] | $\tilde{\mathbf{P}}_{ij} = y_{ij}$ | $\mathbf{C}_{ij} = c - l_{ij}$ | $Q_{ij} = e^{-\mathbf{1}\{i=y\} \cdot C/n_j^{\gamma}}$ | $\epsilon = \tau$ |
| Balanced-Softmax [45] | $\tilde{\mathbf{P}}_{ij} = y_{ij}$ | $\mathbf{C}_{ij} = c - l_{ij}$ | $Q_{ij} = n_j$ | $\epsilon = \tau$ |
| Our Setting | $\tilde{\mathbf{P}}_{ij} = y_{ij}$ | $\mathbf{C}_{ij} = c - l_{ij}$ | $Q = (1 - \lambda(t))\,\text{Uniform} + \lambda(t)\mathbf{r}$ | $\epsilon = \tau$ |

the Balanced Softmax method proposed in [45]. In this paper, motivated by Eq. 10 having a good efficiency in OT and other works [60] which adopt two stages for learning, we let it vary over epochs:

$$\mathbf{Q} = (1 - \lambda(t))\,\text{Uniform} + \lambda(t)\mathbf{r}, \tag{15}$$

where $t$ is the training epoch and $\mathbf{r}$ is the prior probability matrix e.g. the balanced ratio in [45], teacher-based prior predictions or the sample-class-wise setting discussed in Appendix E. $\lambda(t)$ is an epoch varying weight. When $t$ is small, $\lambda(t)$ is close to 0, and $\mathbf{Q}$ is more likely to be a uniform distribution, which gives the model less prior information. As $t$ approaches the final training epoch number $T$, $\lambda(t) \to 1$ and $\mathbf{Q} \to \mathbf{r}$, which means the model is given the full prior information. This gradual process of introducing prior information can be helpful for training the inverse OT. We provide a specific setting for $\lambda(t)$ and $\mathbf{r}$ in Appendix E. During the inference process, the problem reduces to the inner optimization in Eq. 6, which takes the form of traditional unsupervised OT. We use this optimization to obtain the matching between the sample features and labels, which corresponds to classification for the testing data. We do not adopt the long-tailed setting of $\mathbf{Q}$ during inference because the testing data is assumed to be uniform.

### 4.1 Generalization to a Family of Losses in Form of Softmax-based Cross-Entropy

We find our new formulation with inverse OT is a more general form if we set $U = U(\mathbf{a})$ where $\mathbf{a}_i = 1/n$ and many Softmax-based cross-entropy losses can be our special case by varying the form of $\tilde{\mathbf{P}}, \mathbf{C}^{\theta}$ and $\mathbf{Q}$. Specifically, with $U = U(\mathbf{a})$, Eq.14 equals to

$$\min_{\theta} L = \sum_{i,j} \tilde{\mathbf{P}}_{ij} \log \frac{\mathbf{Q}_{ij} e^{-\mathbf{C}_{ij}^{\theta}/\epsilon}}{n \sum_{k=1}^{m} \mathbf{Q}_{ik} e^{-\mathbf{C}_{ik}^{\theta}/\epsilon}}. \tag{16}$$

The proof is given in Appendix F. We can find the form in Eq. 16 is similar with the Softmax cross-entropy loss. In particular, if we set $\tilde{\mathbf{P}}, \mathbf{C}^{\theta}, \mathbf{Q}$ and $\epsilon$ in different forms as shown in Tab. 1, many classification loss can be the special cases of our formulation, as discussed in detail as follows.

**Special cases by varying $\mathbf{C}^{\theta}$.** For a matching problem, it is quite important to define a distance among samples. We first set $\mathbf{C}_{ij}^{\theta} = c - l_{ij}$ where $c$ is a large enough value. In this case, Eq. 16 reduces to a normal softmax-based cross-entropy loss when $\mathbf{Q}$ is in the uniformly distributed. Alternatively, we can set $\mathbf{C}^{\theta}$ differently to suit specific tasks. For example, in learning in the hypersphere space, the Vanilla Softmax will be converted to A-softmax [37] to enforce angular margin constraints for improved face recognition performance.

**Deriving more losses by varying $\mathbf{Q}$.** By varying the setting of $\mathbf{Q}$, we can derive many Softmax-based losses as special cases of the RE-OT-based loss. For example, the Balanced Softmax can be obtained by setting $\mathbf{Q}_{ij} = n_j$, where $n_j$ is the frequency of label $j$ in the training set. This choice of $\mathbf{Q}$ encourages the model to assign equal weights to each class during training, which helps to mitigate the impact of class imbalance.

**Discussion on $\tilde{\mathbf{P}}$.** Re-weighting the example is a common way to improve the Long-tailed learning, including the empirical Re-weighting methods (e.g. focal loss [35]) and automatic Re-weighting ways (e.g. L2RW [46]). In the view of IOT, the supervision of the coupling $\tilde{\mathbf{P}}$ is self adapting and may depend on $\mathbf{P}^{\theta}$, which may be helpful in IOT application fields such as deep graph matching.

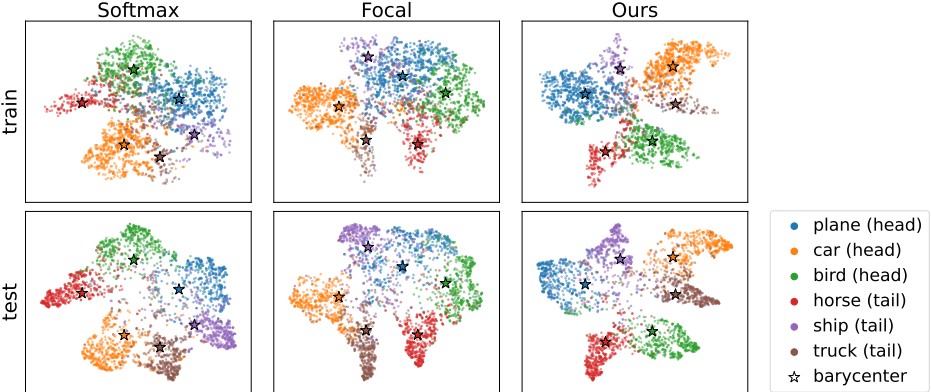

Figure 4: t-SNE for training and testing sample features and class barycenters. The class barycenters are calculated by Eq. 18 with features of training data. It can be observed that in the training data, the class barycenters are mainly on the feature centers of corresponding category samples, which proves the effectiveness of the barycentric projection map defined in Eq. 18. At the same time, we also find that barycenters calculated by features of training samples may not necessarily be centered in the test data. Additionally, the head class occupies a significantly larger space than the tail class in the 2D feature space, which may be one of the reasons for the poor accuracy of the tail class.

**Triplet loss for** $\epsilon \to 0^+$**.** Though Eq. 16 has a very different form compared to triplet loss, the equivalence can also be established when the temperature $\epsilon \to 0^+$. Note that the idea of the relation between triplet loss and our loss is motivated by the work [55] which gives the proof of the equivalence between triplet loss and InfoNCE. We give the proof in Appendix G.

**Representation Learning with our Formulation** Our matching perspective can be applied not only to unbalanced classification but also to unbalanced representation learning, in which we only need to modify the original one-hot label to the augmentation's features. In Sec. 5, we conducted experiments on long-tailed contrastive learning and verified that by selecting an appropriate $\mathbf{Q}$, we can improve the effectiveness of representation learning on long-tailed datasets.

## 4.2 Bridging Regularized OT to Classification

We have shown the classification can be viewed as a matching problem and if we formulate the matching with optimal transport, we can get the equivalence between Softmax (as well as its variants) and RE-OT under the setting of $U(\mathbf{a})$. We can observe that there are some interconnected concepts between OT and classification, which can shed lights on a deeper understanding of OT and in turn the classification problem with different forms. One notable example is the entropy regularization coefficient $\epsilon$ in OT and the temperature coefficient $\tau$ in Softmax. There have been numerous works [24, 61] exploring the role of the temperature hyperparameter in Softmax. We can understand it easier from the view of OT, where $\epsilon$ serves as a relaxation hyperparameter for couplings yielding smoother predictions. In fact, there are other concepts, such as barycentric projection in OT [13], which may help us understand more about classification. We give its definition here:

**Definition 1** (**Barycentric Projection**). *Consider the setting of Eq. 6 in which we use entropic regularization to approximate OT between discrete measures. One can define the so-called barycentric projection map with the coupling* $\mathbf{P}$

$$T : x_i \in X \to \frac{1}{\mathbf{a}_i} \sum_j \mathbf{P}_{ij} y_j \in Y, \tag{17}$$

*where* $\{x_i\} \subset X$ *is the set of locations corresponding to simplex* $\mathbf{a}$ *and* $\{y_j\}$ *is the set of locations in the* $Y$ *space croponding to simplex* $\mathbf{b}$*.*

From the classification view, exactly $T(x_i)$ is probability confidences of samples $x_i$ which can be understand as the feature in the one-hot label space. Besides, motivated by this barycentric projection in OT, we can exactly define the mapping for the feature of one-hot label in hidden feature space:

$$T' : y_j \in Y \to \frac{1}{\sum_i \mathbf{P}_{ij}} \sum_i \mathbf{P}_{ij} \mathbf{f}_i \in \mathcal{F}, \tag{18}$$

where $\mathbf{f}_i$ is the feature of $x_i$ extracting from the neural networks and $\mathcal{F}$ is the feature space. Eq. 18 maps the (one-hot) label to the feature space, which means no need to learn the centroid features

as done in oneline clustering methods [4, 32] but calculate it statically with Eq. 18 if the coupling is known. As shown in Fig. 4, we first train the models using vanilla softmax, focal loss, and our loss on the Cifar10-LT dataset. We consider the logits as the sample features and save the logits from the three head and three tailed classes, along with the corresponding predicted couplings, as matrices. Then, we calculate the barycenters of the labels using Eq. 18, which effectively computes the weighted average of the features. We concat the logits and barycenters as a new matrix and use it to calculate t-SNE results for training data, as shown in the first row of Fig. 4. For testing data, we concat the testing logits with the barycenters calculated from the training data and use t-SNE dimensional reduction, as shown in the second row in Fig. 4. By comparing the first row and the second row of Fig. 4, we can observe the differences in the positions of the barycenters, which are actually caused by shift in the feature distributions.

**Further Discussions between OT and classification.** Note that many theoretical aspects and variations of OT can be incorporated into the field of classification. Here are a few advantages unique to the OT perspective that can be achieved but are not possible in the traditional Bayesian view: **1) Variable** constraints instead of $\{\mathbf{P} : \mathbf{P1} = \mathbf{a}\}$, such as (modified) Optimal Partial Transport (OPT) with $\{\mathbf{P} : \mathbf{P1} \leq \mathbf{a}, \mathbf{1}^\top \mathbf{P1} = s\}$ where $s$ is the number requiring to predict. This allows for distributions that are not just one-vs-all and enables rejection of classification if the model determines that a sample cannot be classified into any of the candidate labels; **2) Generalization** of softmax. The current softmax formulation is essentially based on Entropic Regularization of OT. However, OT regularization goes beyond just the entropic regularization and includes other regularizations such as L2, Tsallis entropies, or divergence-based ones. This opens up possibilities for generalizing softmax; **3) Classification** cross-representation models. From the OT perspective, classification between the features of samples and labels, based on Gromov-Wasserstein Distance, can be performed in different feature spaces. For example, the sample features may be in 100 dimensions while the label features are in 20 dimensions. This can be helpful for preserving privacy as Gromov-Wasserstein Distance [44] only requires similarity between samples and labels. Due to the various variants and theories within OT, considering classification as OT naturally allows us to incorporate existing OT knowledge into the field of classification, which is the ongoing work we are conducting.

# 5 Experiments

We evaluate our proposed approach for both image and molecule datasets. As the primary focus of this paper is the impact of loss functions on model learning, we do not adopt additional specialized techniques to improve performance, such as transfer learning or resampling. This is to control variables and thus all the baselines used in this study represent only the loss functions proposed by them. Additional experimental settings and details can be found in Appendix H.

**Experiments on long-tailed image classification.** The long-tailed image classification datasets analyzed in this study include CIFAR10-LT, CIFAR100-LT [30], and Imagenet-LT [38]. To evaluate the performance of the models, we use the corresponding balanced test dataset and report top-1 accuracy. Specifically, for CIFAR-10, we report accuracy on two sets of classes: Many-shot (more than 100 images) and Few-shot (less than 100 images). For CIFAR-100 and Imagenet, we report accuracy on three sets: Many-shot (more than 100 images), Medium-shot ($20 \sim 100$ images), and Few-shot (less than 20 images). The experiments for unbalanced image classification are all conducted with an imbalanced factor of less than 200, which is defined as the ratio of the number of training instances in the largest class to the smallest [45]. The results for long-tailed classification are presented in Table 3. Note that in our approach, the teacher-based method involves setting the prior matrix $\mathbf{r}$ in Eq. 15 using a vanilla softmax trained teacher model. On the other hand, the ratio-based method refers to setting $\mathbf{r}$ as the long-tailed ratio. From a comprehensive perspective, at least one of our approaches achieves the best average accuracy across the entire dataset.

**Experiments on unbalanced molecule classification.** We further examine the application of our proposed method in the context of molecular representation learning task, which holds significant implications for healthcare [59] and drug discovery [17, 58]. Specifically, we perform the experiments on two unbalanced molecule classification datasets, i.e. OGBG-MOLBBBP and OGBG-MOLBACE, from the Open Graph Benchmark (OGB) [21]. We use the default train/val/test split with ratio 8:1:1. Given the presence of class imbalance within the two datasets, we have chosen to adhere to the precedent set by prior studies, also employing the ROC-AUC as the metric for performance evaluation. As the experimental results are easily influenced by initialization, we conducted five repetitions of

Table 3: Top-1 accuracy (%) for long-tailed image classification with 200 imbalanced factor.

| Method | CIFAR10-LT | | | CIFAR100-LT | | | | ImageNet-LT | | | |
|---|---|---|---|---|---|---|---|---|---|---|---|
| | Many | Few | All | Many | Medium | Few | All | Many | Medium | Few | All |
| Vanilla Softmax | 77.4 | 68.9 | 74.9 | 75.8 | 48.2 | 11.0 | 42.0 | 57.3 | 26.2 | 3.1 | 35.0 |
| Focal Loss [35] | 79.6 | 58.4 | 73.3 | **76.1** | 46.9 | 11.1 | 41.7 | 57.3 | 27.6 | 4.4 | 35.9 |
| LDAM [3] | 80.5 | 65.2 | 75.9 | 75.7 | 50.6 | 11.5 | 42.9 | **57.3** | 27.6 | 4.4 | 35.9 |
| LogitAdjust [39] | 80.0 | 35.3 | 66.6 | 75.7 | 39.2 | 4.1 | 36.5 | 54.2 | 14.0 | 0.4 | 27.6 |
| CB-CE [8] | 76.6 | 70.7 | 74.8 | 53.2 | 48.8 | 13.3 | 36.3 | 35.3 | 32.1 | 21.2 | 31.9 |
| CB-FC [8] | 76.6 | 70.7 | 74.8 | 53.2 | 48.8 | 13.3 | 36.3 | 35.3 | 32.1 | 21.2 | 31.9 |
| Balanced Softmax [45] | 82.2 | 71.6 | 79.0 | 70.3 | 50.4 | 26.5 | 47.0 | 52.5 | 38.6 | 17.8 | 41.1 |
| Ours(teacher-based) | **84.4** | 63.7 | 78.2 | 69.7 | **55.6** | 24.9 | **47.9** | 41.5 | 37.6 | **31.1** | 38.2 |
| Ours(ratio-based) | 81.5 | **74.6** | **79.4** | 70.4 | 53.0 | **26.6** | **47.9** | 53.5 | **39.0** | 17.4 | **41.6** |

the experiment and reported the mean accuracy and its standard deviation in the Tab. 2, which show the effectiveness of our method.

**Experiments on Long-Tailed Instance Segmentation.** We do experiments on long-tailed instance segmentation with LVIS v1.0 datasets [18], as one of the most challenging datasets in vision with a much higher imbalance factor compared to the rest. We use the official splits and evaluation is conducted on validation set. The setting mainly follows the experiments in [45], whose details are given in Appendix H. $AP_m$, $AP_r$, $AP_c$, $AP_f$ are reported in Tab. 2. $AP_m$ denotes Average Precision of masks, $AP_r$, $AP_c$, $AP_f$ denote Average Precision of masks on rare classes, common classes and frequent classes. As shown in Tab. 2, we can find that our method achieves the best $AP_m$ on the entire set and perform competitively for other evaluations.

Table 2: Test on instance segmentation and molecule classification.

| Method | LVIS (instance seg.) | | | | OGBG-MOLBBBP | OGBG-MOLBACE |
|---|---|---|---|---|---|---|
| | $AP_m$ | $AP_r$ | $AP_c$ | $AP_f$ | ROC-AUC | ROC-AUC |
| Vanilla | 20.62 | 9.66 | 19.11 | 27.11 | 69.21 ± 0.34 | 79.40 ± 1.20 |
| Focal Loss [35] | 19.69 | 8.75 | 17.74 | 26.68 | 68.15 ± 1.29 | 80.65 ± 1.50 |
| LDAM [3] | 15.53 | 7.14 | 13.23 | 20.89 | 65.99 ± 0.94 | 79.20 ± 2.00 |
| LogitAdjust [39] | 22.41 | 11.70 | 21.18 | **28.48** | 69.05 ± 1.59 | 81.25 ± 0.33 |
| CB-CE [8] | 20.38 | 9.58 | 18.72 | 26.98 | 69.19 ± 1.18 | 79.39 ± 1.16 |
| CB-FC [8] | 21.41 | **15.67** | 20.34 | 25.25 | 69.51 ± 1.18 | 80.24 ± 1.45 |
| Balanced Softmax | 22.60 | 12.88 | 21.20 | 28.44 | 68.13 ± 0.87 | 80.26 ± 2.28 |
| Ours(ratio-based) | **22.64** | 13.15 | **21.26** | 28.34 | **70.48 ± 0.73** | **82.48 ± 1.59** |

**More Experiments.** We also perform experiments on Long-Tailed contrastive representation learning on CIFAR10-LT, CIFAR100-LT with 100 imbalanced factor. We compare our methods with the methods in [5, 27]. The settings and results are given in Appendix H.5. Besides, to further demonstrate the usefulness of viewing classification as OT, a new inference method is proposed for testing. we replaced the softmax (i.e. constraints within $U(\mathbf{a})$) with Sinkhorn algorithm (i.e. constraints within $U(\mathbf{a}, \mathbf{b})$), where $\mathbf{b}$ represents the assumed ratio in the testing data (e.g., long-tailed, uniform, or reverse long-tailed distribution). Details are given in Appendix H.6.

## 6 Conclusion

We have provided a matching perspective to provide new understanding to classification. Under our matching-based framework, we show that the inherent challenge of long-tailed classification lies in the inconsistency of the prior information used for training and testing under the matching formulation. We then develop inverse relative entropic OT approach to revisit the classification problem, and especially for the long-tailed case, we develop a simple yet effective technique based on our theoretical insights. Experimental results verify the effectiveness of our methods.

**Broader impacts.** Our work establishes a theoretical framework for understanding and solving (self-)supervised problems using OT theory. This not only broadens the range of applications for OT in generative models, such as WGAN and its variants, but also provides a new perspective for redefining, understanding, and solving (self-)supervised problems. Furthermore, the theory and concepts of OT can facilitate the discovery of interesting relationships and insights for supervised problems, such as the connection between open set recognition and unbalanced OT. **Limitations.** It assumes known prior label distribution testing data, which is often unknown in real-world scenarios.

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
