# Appendix

## A The Proof of Prop. 1

Denote $\mathbf{P}_Q^\epsilon$ and $\mathbf{P}^\epsilon$ as the optimal solution of RE-OT and entropic regularized OT, respectively. Then we can get the following two results.

**1)** When $\epsilon \to +\infty$, the RE-OT's optimal solution $\mathbf{P}_Q^\epsilon$ will converge to $\widetilde{\mathbf{Q}}$ where $\widetilde{\mathbf{Q}}$ has the form $\widetilde{\mathbf{Q}} = \mathrm{diag}(\mathbf{u}')\mathbf{Q}\,\mathrm{diag}(\mathbf{v}')$ with two uniquely defined non-negative vectors $\mathbf{u}'$ and $\mathbf{v}'$.

*Proof.* While $\epsilon \to +\infty$, one should rather consider the problem

$$\min_{\mathbf{P}\in U(\mathbf{a},\mathbf{b})} -H_Q(\mathbf{P}) = \sum_{ij} \mathbf{P}_{ij}\left(\log\left(\frac{\mathbf{P}_{ij}}{\mathbf{Q}_{ij}}\right) - 1\right). \tag{19}$$

We can use the Lagrange methods

$$\mathcal{L} = -H_Q(\mathbf{P}) - <\mathbf{f}', \mathbf{P}\mathbf{1}_m - \mathbf{a}> - <\mathbf{g}', \mathbf{P}^\top\mathbf{1}_n - \mathbf{b}> \tag{20}$$

Then we can get

$$\frac{\partial L(\mathbf{P})}{\partial \mathbf{P}_{ij}} = (\log\mathbf{P}_{ij} - \log\mathbf{Q}_{ij}) - \mathbf{f}'_i - \mathbf{g}'_j = 0. \tag{21}$$

Then we have $\mathbf{P}_{ij} = e^{\mathbf{f}'_i}\mathbf{Q}_{ij}e^{\mathbf{g}'_j}$. By setting $\mathbf{u}' = e^{\mathbf{f}'}$ and $\mathbf{v}' = e^{\mathbf{g}'}$, we have the optimal solution $\widetilde{\mathbf{Q}} = \mathbf{P} = \mathrm{diag}(\mathbf{u}')\mathbf{Q}\,\mathrm{diag}(\mathbf{v}')$. So $\widetilde{\mathbf{Q}}$ can be calculated by the iterations of row normalization to $\mathbf{a}$ and column normalization to $\mathbf{b}$ from $\mathbf{Q}$. Specially, when $Q = \mathbf{1}_{n\times m}$, we find $\mathbf{u}' = \mathbf{a}$ and $\mathbf{v}' = \mathbf{b}$ is one of the solution for Eq. 3 under $U(\mathbf{a}, \mathbf{b})$. And due to the convex of the objective function, the solution is unique thus $\widetilde{\mathbf{Q}} = \mathrm{diag}(\mathbf{u}')\mathbf{1}_{n\times m}\,\mathrm{diag}(\mathbf{v}') = \mathbf{a} \otimes \mathbf{b}$ □

**2)** With the prior $\mathbf{Q}$ and its corresponding $\widetilde{\mathbf{Q}}$ as defined in (1), we have $\mathbf{P}_Q^\epsilon = \mathbf{P}_{\widetilde{Q}}^\epsilon$, and when $\widetilde{\mathbf{Q}} = \mathbf{a} \otimes \mathbf{b}$, we have $\mathbf{P}^\epsilon = \mathbf{P}_{\widetilde{Q}}^\epsilon$.

*Proof.* We first get the optimal solution $\mathbf{P}_Q^\epsilon$. Consider the problem

$$\min_{\mathbf{P}\in U(\mathbf{a},\mathbf{b})} <\mathbf{C}, \mathbf{P}> -\epsilon H_Q(\mathbf{P}). \tag{22}$$

We can use the Lagrange methods:

$$\mathcal{L} = <\mathbf{C}, \mathbf{P}> -\epsilon H_Q(\mathbf{P}) - <\mathbf{f}, \mathbf{P}\mathbf{1}_m - \mathbf{a}> - <\mathbf{g}, \mathbf{P}^\top\mathbf{1}_n - \mathbf{b}>. \tag{23}$$

Then we can get

$$\frac{\partial L(\mathbf{P})}{\partial \mathbf{P}_{ij}} = \mathbf{C}_{ij} + \epsilon\left(\log\mathbf{P}_{ij} - \log\mathbf{Q}_{ij}\right) - \mathbf{f}_i - \mathbf{g}_j = 0. \tag{24}$$

So we have the solution form

$$(\mathbf{P}_Q^\epsilon)_{ij} = e^{\mathbf{f}_i/\epsilon}\mathbf{Q}_{ij}e^{(-\mathbf{C}_{ij}/\epsilon)}e^{\mathbf{g}_j/\epsilon}. \tag{25}$$

And thus we have the $\mathbf{P}_Q^\epsilon = \mathrm{diag}(\mathbf{u})\mathbf{Q}\,\mathrm{diag}(\mathbf{v})$ where $\mathbf{u} = e^{\mathbf{f}/\epsilon}$ and $\mathbf{v} = e^{\mathbf{g}/\epsilon}$.

Similarly, for the optimization

$$\min_{\mathbf{P}\in U(\mathbf{a},\mathbf{b})} <\mathbf{C}, \mathbf{P}> -\epsilon H_{\widetilde{\mathbf{Q}}}(\mathbf{P}), \tag{26}$$

we also have the solution form

$$(\mathbf{P}_{\widetilde{\mathbf{Q}}}^\epsilon)_{ij} = e^{\widetilde{\mathbf{f}}_i/\epsilon}\widetilde{\mathbf{Q}}_{ij}e^{(-\mathbf{C}_{ij}/\epsilon)}e^{\widetilde{\mathbf{g}}_j/\epsilon}. \tag{27}$$

According the Proof 1, we have

$$\tilde{\mathbf{Q}}_{ij} = e^{\mathbf{f}'_i/\epsilon}\mathbf{Q}_{ij}e^{\mathbf{g}'_j/\epsilon} \tag{28}$$

With Eq. 27 and Eq. 28, we have

$$
\begin{aligned}
(\mathbf{P}^\epsilon_{\widetilde{\mathbf{Q}}})_{ij} &= e^{\widetilde{\mathbf{f}}_i/\epsilon} e^{\mathbf{f}'_i} \mathbf{Q}_{ij} e^{(-\mathbf{C}_{ij}/\epsilon)} e^{\mathbf{g}'_j} e^{\widetilde{\mathbf{g}}_j/\epsilon} \\
&= e^{\widetilde{\mathbf{f}}_i/\epsilon + \mathbf{f}'_i} \mathbf{Q}_{ij} e^{(-\mathbf{C}_{ij}/\epsilon)} e^{\widetilde{g}_j/\epsilon + \mathbf{g}'_j}.
\end{aligned}
\tag{29}
$$

Compared with Eq. 25 and Eq. 29, we find $\mathbf{P}^\epsilon_{\widetilde{\mathbf{Q}}}$ and $\mathbf{P}^\epsilon_{\mathbf{Q}}$ are both the projection of $\mathbf{K} = \mathbf{Q}_{ij} e^{(-\mathbf{C}_{ij}/\epsilon)}$ onto $U(\mathbf{a}, \mathbf{b})$ and thus have the same solutions.

When $\mathbf{Q} = \mathbf{1}_{n \times m}$, we can get $\mathbf{P}^\epsilon_{\mathbf{Q}} = \mathbf{P}^\epsilon$. As discussed in Proof 1, we know that $\tilde{\mathbf{Q}} = \mathbf{a} \times \mathbf{b}$. Then we can get

$$
\mathbf{P}^\epsilon = \mathbf{P}^\epsilon_{\mathbf{Q}} = \mathbf{P}^\epsilon_{\tilde{\mathbf{Q}}}
\tag{30}
$$

for $\mathbf{Q} = \mathbf{1}_{n \times m}$ and $\tilde{\mathbf{Q}} = \mathbf{a} \times \mathbf{b}$. Fig. 5 is the toy experiment with histograms and cost matrix in the first row. The second and third row are Sinkhorn results of $\mathbf{P}_{\mathbf{Q}^\epsilon}$ and $\mathbf{P}_{\tilde{\mathbf{Q}}^\epsilon}$ when $\epsilon = 0.01, 0.1, 1$ and 10. We can see the approximate solutions. $\qquad\square$

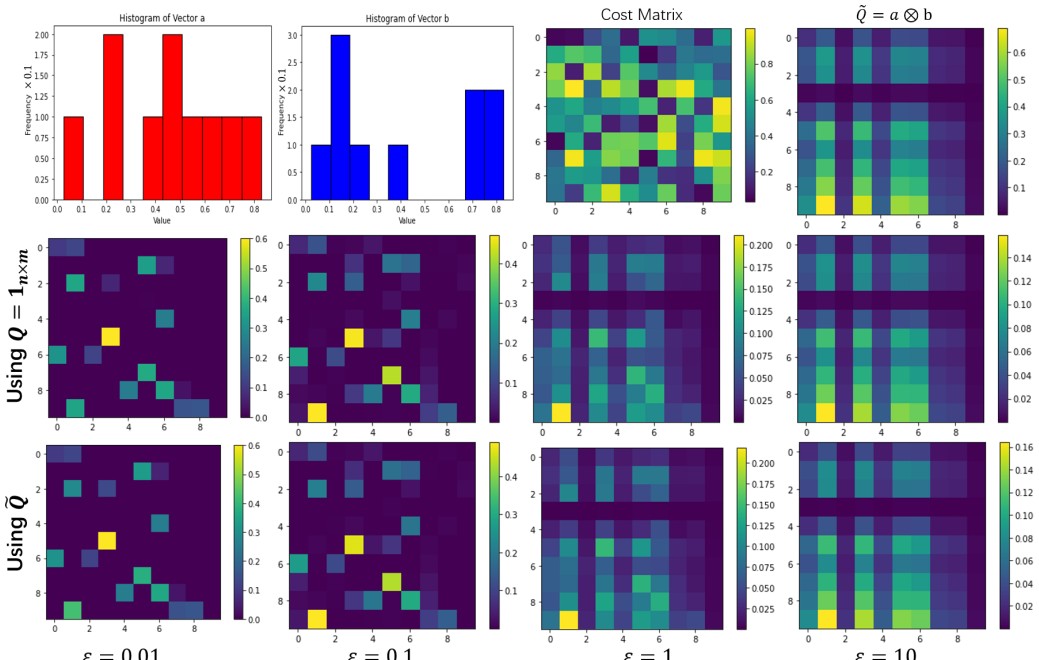

Figure 5: The results of RE-OT $\mathbf{P}^\epsilon_{\mathbf{Q}}$ and $\mathbf{P}^\epsilon_{\tilde{\mathbf{Q}}}$ by varying $\epsilon = 0.01, 0.1, 1$ and 10 where $\mathbf{Q}$ is set to $\mathbf{1}_{n \times m}$.

## B    The proof of Prop. 2

*Proof.* From the definition of $\widetilde{KL}$ and $\mathbf{K}_{ij} = \mathbf{Q}_{ij} e^{-\mathbf{C}_{ij}/\epsilon}$, we have

$$
\begin{aligned}
\min_{\mathbf{P} \in U(\mathbf{a},\mathbf{b})} \widetilde{KL}(\mathbf{P}|\mathbf{K}) &=_{\mathbf{P} \in U(\mathbf{a},\mathbf{b})} \sum_{ij} \left( \mathbf{P}_{ij} \log \mathbf{P}_{ij} - \mathbf{P}_{ij} - \mathbf{P}_{ij} \log \mathbf{Q}_{ij} - \mathbf{P}_{ij} \log e^{-\mathbf{C}_{ij}/\epsilon} \right) \\
&= \min_{\mathbf{P} \in U(\mathbf{a},\mathbf{b})} \sum_{ij} \left( \mathbf{P}_{ij} \left( \log \frac{\mathbf{P}_{ij}}{\mathbf{Q}_{ij}} - 1 \right) + \frac{1}{\epsilon} \mathbf{P}_{ij} \mathbf{C}_{ij} \right) \\
&= \min_{\mathbf{P} \in U(\mathbf{a},\mathbf{b})} \frac{1}{\epsilon} L^\epsilon_Q.
\end{aligned}
\tag{31}
$$

So the optimizations in Eq. 6 and Eq. 8 are equal. $\qquad\square$

## C    The Sinkhron Algorithm and its proof for RE-OT

As discussed in Appendix A, we have $\mathbf{P} = diag(e^{\mathbf{f}/\epsilon})\mathbf{K}diag(e^{\mathbf{g}/\epsilon}) \in U(\mathbf{a}, \mathbf{b})$. Due to $\mathbf{P1}_m = \mathbf{a}$ and $\mathbf{P}^\top\mathbf{1}_n = \mathbf{b}$, we have

$$\mathbf{u} \odot \mathbf{Kv} = \mathbf{a} \quad \text{and} \quad \mathbf{v} \odot \mathbf{K}^\top \mathbf{u} = \mathbf{b}, \tag{32}$$

so we can update $\mathbf{u}$ and $\mathbf{v}$ by

$$\mathbf{u}^{(l+1)} = \frac{\mathbf{a}}{\mathbf{Kv}^{(l)}} \quad \text{and} \quad \mathbf{v}^{(l+1)} = \frac{\mathbf{a}}{\mathbf{K}^\top\mathbf{v}^{(l+1)}}, \tag{33}$$

initialized with an arbitrary positive vector $\mathbf{v}^{(0)} = \mathbf{1}_n$. The algorithm is shown below.

---
**Algorithm 1** Sinkhorn Algorithm for RE-OT
---
**Input:** Cost matrix $C$, histograms $\mathbf{a}$ and $\mathbf{b}$, Prior $\mathbf{Q}$
**Output:** $P$
  Initialize $\mathbf{K} = \mathbf{Q} \odot e^{-\mathbf{C}/\epsilon}$ and $l = 0$, $\mathbf{v}^{(l)} = \mathbf{1}_n$
  **while** $\mathbf{v}^{(l)}$ and $\mathbf{u}^{(l)}$ not converge **do**
    $\mathbf{u}^{(l+1)} = \frac{\mathbf{a}}{\mathbf{Kv}^{(l)}}$
    $\mathbf{v}^{(l+1)} = \frac{\mathbf{a}}{\mathbf{K}^\top\mathbf{v}^{(l+1)}}$
    $l+ = 1$
  **end while**
  **return** $P = diag(\mathbf{u}^{(l)})\mathbf{K}diag(\mathbf{v}^{(l)})$

---

## D    Dual for RE-OT

*Proof.* As discussed in Appendix A, we have $\mathbf{P}_{ij} = e^{\mathbf{f}_i/\epsilon}\mathbf{K}_{ij}e^{\mathbf{g}_j/\epsilon} = e^{\mathbf{f}_i/\epsilon}\mathbf{Q}_{ij}e^{-\mathbf{C}_{ij}/\epsilon}e^{\mathbf{g}_j/\epsilon}$. Then the optimization in Eq. 6 is equal to

$$\mathcal{L}_{\mathbf{Q}}^\epsilon = <e^{\mathbf{f}/\epsilon}, \mathbf{K} \odot \mathbf{C}e^{\mathbf{g}/\epsilon}> -\epsilon H_{\mathbf{Q}}(diag(e^{\mathbf{f}/\epsilon})\mathbf{K}diag(e^{\mathbf{g}/\epsilon})). \tag{34}$$

For the regularization term $\epsilon H_{\mathbf{Q}}(\mathbf{P}) = \epsilon < \mathbf{P}, \log(\mathbf{P}) - \log(\mathbf{Q}) - \mathbf{1}_{n\times m} >$, we have

$$\epsilon H_{\mathbf{Q}}(\mathbf{P}) = \mathbf{f}^\top(\mathbf{P1}_m - \mathbf{a}) - \mathbf{g}^\top(\mathbf{P}^\top\mathbf{1}_n - \mathbf{b})$$
$$= <diag(e^{\mathbf{f}/\epsilon})\mathbf{K}diag(e^{\mathbf{g}/\epsilon}), \mathbf{f1}_m^\top + \epsilon\log\mathbf{Q} + \mathbf{1}_n\mathbf{g}^\top - \mathbf{C} - \epsilon\log\mathbf{Q} - \epsilon\mathbf{1}_{n\times m}> \tag{35}$$
$$= <e^{\mathbf{f}/\epsilon}, \mathbf{K} \odot \mathbf{C}e^{\mathbf{g}/\epsilon}> + <\mathbf{f}, \mathbf{a}> + <\mathbf{g}, \mathbf{b}> -\epsilon < e^{\mathbf{f}/\epsilon}, \mathbf{K}e^{\mathbf{g}/\epsilon}>,$$

where $\odot$ is the element-wise product. So the optimization equals to

$$\max_{\mathbf{f}\in R_n, \mathbf{g}\in R_m} <\mathbf{f}, \mathbf{a}> + <\mathbf{g}, \mathbf{b}> -\epsilon < e^{\mathbf{f}/\epsilon}, \mathbf{K}e^{\mathbf{g}/\epsilon}>, \tag{36}$$

which is the dual formulation. $\qquad\square$

## E    The setting for Q in Long-tailed Learnig

We set $\mathbf{Q}$ by varying over epochs:

$$\mathbf{Q} = (1 - \lambda(t)) \text{Uniform} + \lambda(t)\mathbf{r}, \tag{37}$$

where we let $\lambda(t)$ be a piece-wise linear function:

$$\lambda(t) = \begin{cases} 0 & \text{if } t < t_1 \\ \frac{t-t_1}{t_2-t} & \text{if } t_1 < t < t_2 \\ 1 & \text{if } t_2 < t < T \end{cases}. \tag{38}$$

Here $t$ is the training epoch number and $t_1, t_2, t_3$ are hyper-parameters. And for the setting of $\mathbf{r}$, a simple set is the balanced ratio as used in [45] which is specified as

$$\mathbf{r}_{ij} = \frac{n_j}{\sum_k n_k}, \tag{39}$$

where $n_j$ is the number of samples in $j$ class. Or we can set $\mathbf{r}$ as

$$\mathbf{r}_{ij} = \frac{n_j^b}{n} \tag{40}$$

where $n_j^b$ is the mini-bath samples' number in $j$ class and $n$ is the batchsize. We can also adopt the interpolation of Eq. 39 and Eq. 40, which is specified as

$$\mathbf{r} = \gamma \times \frac{n_j}{\sum_k n_k} + (1 - \gamma) \times \frac{n_j^b}{n}, \tag{41}$$

where $\gamma$ is a smoothing parameter determining the degree of blending between Eq. 40 and Eq. 39. The above three are all class-wise and we can also define $\mathbf{r}$ with class-sample-wise form with

$$\mathbf{r}_{ij} = \frac{e^{l_{ij}}}{\sum_k e^{l_{ik}}} * \frac{n_j}{\sum_k n_k}, \tag{42}$$

where $\frac{e^{l_{ij}}}{\sum_k e^{l_{ik}}}$ is used to control the simple-wise factor with different sample confidences.

## F    Proof when P in U(a)

Now we show the softmax with the constraints:

$$U(\mathbf{a}) = \{\mathbf{P} \in \mathbb{R}_+^{n \times m} | \mathbf{P} \mathbf{1}_m = \mathbf{a}\} \tag{43}$$

where $\mathbf{a} = \mathbf{1}/n$ and $\mathbf{1}_m$ is the $m$-dimensional column vector whose elements are all ones. With the objective of the RE-OT:

$$\mathbf{P}^\theta = \arg \min_{P \in U(\mathbf{a})} < \mathbf{C}^\theta, \mathbf{P} > -\epsilon H_{\mathbf{Q}}(\mathbf{P}), \tag{44}$$

We introduce the dual variable $\mathbf{f} \in R^n$. The Lagrangian of the above equation is:

$$L(\mathbf{P}, \mathbf{f}) = < \mathbf{C}, \mathbf{P} > -\epsilon H_{\mathbf{Q}}(\mathbf{P}) - \sum_{i=1}^n \mathbf{f}_i \cdot \left( \sum_{j=1}^m \mathbf{P}_{ij} - \frac{1}{n} \right) \tag{45}$$

The first order conditions then yield by:

$$\frac{\partial L(\mathbf{P}, \mathbf{f})}{\partial \mathbf{P}_{ij}} = \mathbf{C}_{ij} + \epsilon(\log \mathbf{P}_{ij} - \log \mathbf{Q}_{ij}) - \mathbf{f}_i = 0. \tag{46}$$

Thus we have $\mathbf{P_{ij}} = \mathbf{Q}_{ij} e^{(\mathbf{f}_i - C_{ij}^\theta)/\epsilon}$ for every $i$ and $j$, for optimal $\mathbf{P}$ coupling to the regularized problem. Due to $\sum_j \mathbf{P}_{ij} = 1/n$ for every $i$, we can calculate the Lagrangian parameter $\mathbf{f}_i$ and the solution of the coupling is given by:

$$\mathbf{P}_{ij} = \frac{\mathbf{Q}_{ij} \exp\left(-\mathbf{C}_{ij}^\theta/\epsilon\right)}{n \sum_{k=1}^m \mathbf{Q}_{ik} \exp\left(-\mathbf{C}_{it}^\theta/\epsilon\right)}. \tag{47}$$

Then in outer minimization, if we set $\tilde{P}_{ij} = y_{ij}$, the optimization in Eq. 14 is equal to

$$\mathcal{L} = -\sum_{i=1}^n \log \left( \frac{\mathbf{Q}_{ij} \exp\left(-\mathbf{C}_{ij}^\theta/\epsilon\right)}{n \sum_{k=1}^m \mathbf{Q}_{ik} \exp\left(-\mathbf{C}_{ik}^\theta/\epsilon\right)} \right). \tag{48}$$

## G    The triplet loss

When $\epsilon \to 0$, we can find that

$$\begin{aligned}
&\lim_{\epsilon \to 0^+} -\log \frac{\exp(-C_{ij}/\epsilon)}{\exp(-C_{ij}/\epsilon) + \sum_{k \neq j} \exp(-C_{ik}/\epsilon)} \\
&= \lim_{\epsilon \to 0^+} +\log(1 + \sum_{k \neq j} \exp(C_{ij} - C_{ik}/\epsilon)) \\
&\approx \lim_{\epsilon \to 0^+} +\log(1 + \sum_{C_{ij} > C_{ik}} \exp(C_{ij} - C_{ik}/\epsilon)) \\
&\approx \frac{1}{\epsilon} \lim_{\epsilon \to 0^+} \min\{C_{ij} - \max_k C_{ij}, 0\}
\end{aligned} \tag{49}$$

Table 4: Top-1 accuracy (%) for long-tailed image classification with 10/100 imbalanced factor on CIFAR10-LT and CIFAR100-LT.

| Method | CIFAR10-LT | | CIFAR100-LT | |
|---|---|---|---|---|
| | IF=10 | IF=100 | IF=10 | IF=100 |
| Vanilla Softmax | 90.53 | 79.12 | 62.51 | 46.00 |
| Focal Loss [35] | 89.61 | 78.02 | 61.77 | 45.41 |
| LDAM [3] | 89.88 | 80.13 | 58.28 | 46.84 |
| LogitAdjust [39] | 88.85 | 73.71 | 58.89 | 39.61 |
| CB-CE [8] | 90.30 | 79.40 | 61.99 | 42.96 |
| CB-FC [8] | 90.37 | 79.31 | 62.43 | 42.57 |
| Balanced Softmax [45] | 91.00 | 82.85 | 64.59 | 61.57 |
| Ours | 91.83 | 83.79 | 64.65 | 51.79 |

So with a feature extractor $f$, if we define $y_{ij} = 1$ if $x_i$ and $x_j$ are positive pair and $y_{ij} = 0$ otherwise, we can get the triplet loss from Eq. 16 that

$$\sum_{ij} y_{ij} \max\{||f(x_i) - f(x_j)||^2 - \min_k ||f(x_i) - f(x_k)||^2, 0\} \tag{50}$$

where $\tilde{\mathbf{P}}_{ij} = y_{ij}$ and $C_{ij} = ||f(x_i) - f(x_j)||^2$ as the setting given in Tab. 1.

# H   More Experimental Setting and Results

## H.1   Hardware and Software

We use Intel Core i9-10920X CPU @ 3.50GHz with Nvidia GeForce RTX 3090 GPU for model training. We take single GPU to train models on CIFAR-10-LT, CIRFAR-100-LT, ImageNet-LT, OGBG-MOLBBBP and OGBG-MOLBACE and 8 GPUs to train models on LVIS. We implement our proposed algorithm with PyTorch-1.4.0 for all experiments.

## H.2   More Experimental Setting Details and Results for Image Classification

We perform the long-tailed image classification task on CIFAR10-LT, CIFAR100-LT [30], and Imagenet-LT [38] datasets, and evaluate on balanced testing data by reporting top-1 accuracy. For CIFAR10 and CIFAR100, the experiments of image classificaiton tasks are based on ResNet32 [20] as the backbone with 0.05 learning rate, while for Imagenet dataset, we use ResNet10 for training with 0.2 learning rate. For the setting of $\mathbf{Q}$, we adopt the Eq. 41 form where We train all the data with 15000 iterations on a single GPU and imbalanced ratio is set as 200, 100, 10, resp. The experimental results are shown in Tab. 3 and Tab. 4.

## H.3   More Setting Details for Unbalanced Molecule Classification

The molecule experiments are done in OGBG-MOLBBBP and OGBG-MILLRACE datasets from the Open Graph Benchmark (OGB) [21], which are widely used in the molecular representation learning field [17, 58]. The tasks are binary classification. OGBG-MOLBBBP is a dataset of Brain-Blood Barrier Penetration and each molecule has a label indicating whether it can penetrate through brain cell membrane to enter central nervous system. While OGBG-MILLRACE is a dataset of blinding affinity against hunmanbeta-screatas. Label 1 represents the molecule can blind to human beta-secretase 1. We use the default train/val/test split with ratio 8:1:1. In detail, OGBG-MOLBBBP has $84.20\%$ label 1 molecules for training, and $40.69, 52.94$ for validation and testing. And for OGBG-MOLBACE data, it has $84.20\%$, $86.09\%$ and $53.29\%$ label 1 data for training, validation and testing data splits. Given the presence of class imbalance within the two datasets, we have chosen to adhere to the precedent set by prior studies, also employing the ROC-AUC as the metric

for performance evaluation. The experiments are easily influenced by the parameter initialization. So we repeat 5 times and report the mean and std in Tab. 2. We adopt 5 layer GCN with 0.1 drop ratio and output 256 embedding dimensions to binary classification with linear projection. The batch size was set to 32 with 0.001 learning rate. The AUC-ROC results are reported in Tab. 2.

## H.4 More Setting Details for on Long-Tailed Instance Segmentation

The experiments of Instance Segmentation are based on LVIS v1.0 datasets [18], which is one of the most challenging datasets in vision with a much higher imbalance factor compared to the rest. We use the official splits and evaluation is conducted on validation set. The setting mainly follows the experiments in [45]. We use the off-the-shelf model Mask R-CNN with the backbone network ResNet-50 for LVIS, which is pre-trained on ImageNet. We use an SGD optimizer with 0.9 momentum, 0.02 initial learning rate, and 0.0001 weight decay. The model is trained for 22k iterations with 8 images per mini-batch. The learning rate is dropped by a factor of 10 at both 11k iterations and 18k iterations. The bounding box classifier consists of one fully connected layer. In our method, $\lambda(t)$ in this case is set to be the constant 0.1 and $\gamma$ is set to be 0.5 in Eq. 41 and finally the matrix $\mathbf{r}$ is normalized for training.

## H.5 Experiments Setting and Results for Supervised Contrastive Learning

The experiments of Long-Tailed Contrastive Learning are performed on CIFAR10-LT, CIFAR100-LT datasets and imbalanced ratio is set as 100. We use ResNet-50 as the encoder backbone. The augmentation includes random cropping, horizontal flipping, color distortions and gray scale. The architecture of network and training strategy mainly follow [27]. In the pertaining stage, we use an SGD optimizer with 0.5 initial learning rate, 0.0001 weight decay, 0.9 momentum. The models are trained for 500 epochs with a mini batch size of 512. The learning rate adjustment strategy follows [27]. The temperature is set to be 0.1 for all contrastive losses. Then we evaluate the performance of our pretrained model using linear classification and k-nearest neighbors classification. With all convolutional layers frozen, we only train the last linear classifier using the SGD optimizer with 1 initial learning rate and 0.9 momentum for 100 epochs with a mini batch size of 512. We also use the same learning rate adjustment strategy as [27]. We also compare k-nearest neighbors classifier (k-NN,k= 5 here) with linear evaluation. We define the cost matrix as $C_{ij} = c - z_i \cdot z'_j$ where $z_i$ and $z'_j$ are the normalized features of sample $i$ and feature of augmentational sample $j$. From Eq. 16, the contrastive loss is specified as

$$L_{constrast} = -\sum_{i \in B} \frac{1}{|P(i)|} \sum_{j \in P(i)} \log \frac{Q_{ik} \exp(z_i \cdot z'_j/\epsilon)}{\sum_{j \in B, k \neq i} Q_{ik} \exp(z_i \cdot z'_k/\epsilon)}, \tag{51}$$

where $B$ is the batch for constrastive learning as same in [5], $P(i)$ is the collection where the samples have the same label as the sample $i$. For the matrix $Q$, $\lambda(t)$ in this case is set to be a constant and $\mathbf{r}$ is defined as follows:

$$\mathbf{r_{ij}} = \frac{R_i * R_j}{\max\{R_m * R_n\}_{m,n}}, \quad \text{where} \quad R_k = \gamma \times \frac{n_j}{\sum_k n_k} + (1-\gamma) \times \frac{n_k^b}{n}. \tag{52}$$

Here, $n_j$ and $n_j^b$ are defined in Eq. 41 and $\gamma$ is set as 0.5. We show the experiments in the following table.

Table 5: Comparison on CIFAR10/100-LT with constrastive methods.Top-1 accuracy (%) is reported with 100 imbalanced ratio.

| Method | CIFAR10-LT | CIFAR100-LT |
|---|---|---|
| SimCLR [5] | 56.70 | 29.27 |
| SupCon [27] | 72.60 | **41.16** |
| Ours | **73.94** | 40.92 |

## H.6 Sinkhorn Inference with a trained model

We conducted a simple experiment to further demonstrate the usefulness of viewing classification as OT. In the inference phase on the testing data, we replaced the softmax (i.e. constraints within $U(\mathbf{a})$)

with Sinkhorn algorithm (i.e. constraints within $U(\mathbf{a}, \mathbf{b})$), where $\mathbf{b}$ represents the assumed ratio in the testing data (e.g., long-tailed, uniform, or reverse long-tailed distribution). The results are shown below:

Table 6: Top-1 accuracy (%) classification with long-tailed, uniform and reverse long-tailed testing data given the vanilla softmax and our loss trained models.

| Method | Model trained with Vanilla Softmax | | | Model trained with our loss | | |
|---|---|---|---|---|---|---|
| | LT | Uniform | Reverse LT | LT | Uniform | Reverse LT |
| Inference with Softmax | 58.7 | 40.8 | 23.5 | 60.4 | 47.4 | 35.1 |
| Inference with Sinkhorn | 59.1 | 46.6 | 39.6 | 60.4 | 47.9 | 40.9 |

In the above experiments, we reconfigured the testing data to have long-tailed (IF=10), uniform (unchanged), and reverse long-tailed (IF=10) distributions. We found that using Sinkhorn as the inference method during the testing process can lead to significant improvements, particularly for models trained with vanilla softmax. These experiments validate the effectiveness of treating classification as Optimal Transport during the testing inference stage.