# OpenReview forum: "Relative Entropic Optimal Transport: a (Prior-aware) Matching Perspective to (Unbalanced) Classification"
_NeurIPS.cc/2023/Conference — NeurIPS 2023 poster_

### Official Review · Reviewer_HboX · 2023-06-22

**Soundness:** 4 excellent
**Presentation:** 2 fair
**Contribution:** 2 fair
**Rating:** 5
**Confidence:** 2

**Summary:**

The paper proposes an inverse Relative Entropic Optimal Transport (RE-OT) point of view for classification problems. The paper then proposes to use inverse RE-OT with a time-varying prior for solving long-tailed classification problems. Evaluations show improvements compared to existing baselines such as vanilla softmax and balanced softmax on long-tailed tasks.

**Strengths:**

The reviewer believes the main strengths of the paper are solid theory and novelty of the framework. Specifically,

- The OT and novel RE-OT theory is introduced clearly, with provided visualizations and explanations to aid understanding.
- The novel RE-OT theory makes a nice connection and a unified view for many existing classification methods, as summarized in Table 1.
- Experiment results show some advantage compared to existing methods.


**Weaknesses:**

The reviewer mainly has concerns with the impact of the work. In particular,

- Some additional intuitions would be helpful for the reviewer on why RE-OT gives a good or useful way for tackling classification problems. Currently, the RE-OT framework is laid out clearly, and the authors proposed to apply RE-OT to solve long-tailed classification, but the paper does not make clear what methodological or practical benefit the proposed framework has.
- Comparing different methods in Table 1, the newly proposed method only differs from Balanced-Softmax in the choice of Q slightly, and the proposed choice of Q is not properly motivated in my opinion. The work also does not compare with using epoch-varying losses or two-stage approaches for other methods.
- The evaluations show relatively minor improvements compared to existing works, and most experiments do not report error bars so it is unclear if most improvements are significant.

The work also has some minor issues with clarity:
- The work uses $n$ for both number of iterations and dimension of polytope $\Sigma_n$ which can be confusing.
- Proposition 3 appears to be misplaced.
- The notations in Figure 2 and equation (10) are quite hard to understand.


**Questions:**

- How does (15) relate to (10)?
- Are (17) and (18) unique to the OT framework?

**Limitations:**

The authors have addressed one of the limitations of the work, i.e. the assumption of known label distribution of test data. The authors have not addressed potential negative societal impacts of their work as it is mostly theoretical and methodological.

---

> ### Author Rebuttal · Authors · 2023-08-09
>
> We truly appreciate the time and effort you have spent reviewing this paper. Below are our responses:
>
> >***Q1: the paper does not make clear what methodological or practical benefit the proposed framework has.***
>
> We think that our (RE-)OT framework for classification is highly motivated and valuable. Although we mentioned in the future work section that the OT perspective might help approach open-set problems, we can elaborate more on this direction if you are willing to hear about it. Here are a few directions we can do from the OT perspective yet less possible in the traditional Bayesian way:
>
> **1.  Change in the coupling constraint  $\\{ P: P1=a \\}$**.  We can adopt other constraints instead of $\\{ P: P1=a \\}$, such as using the constraints in (modified) Optimal Partial Transport (OPT) with $\\{ P: P1 \leq a,1^\top P1=s \\}$. This allows the model that is not just choosing one from all classes and enables rejection of classification if the model determines that a sample cannot be classified into any of the candidate labels.
>
> **2. Change in label selection in one model**.  In the OT perspective, classification is not understood in terms of $P(y|x)$ but rather from a matching perspective. Therefore, when we have representations of samples and labels, we can freely choose different label sets to form different classification tasks. For example, during training, we classify into ${y_1,y_2,\dots,y_{10}}$, but during testing, we classify into ${y_3,y_7,y_{10}}$. Additionally, if the label representation is based on NLP embedding models, we can include previously unseen labels in the classification testing phase, potentially addressing zero-shot learning problems without the need for additional parameter training.
>
> You can see more in the A3 response of Reviewer RCtv (due to the length limitation).
>
> Because of the various variants and theories within OT, considering classification as OT naturally allows us to incorporate existing OT knowledge into the field of classification, which is the ongoing work we are conducting. Only through this OT perspective, it becomes possible in the future to develop models that can "Classify Anything," including different task selections, variable classification constraints, and more in one model. Therefore, the main purpose of this paper is to provide a different perspective on classification models and offer conceptual assistance to general recognition models.
>
> >***Q2: "Comparing different methods in Table 1, the newly proposed method only differs from Balanced-Softmax in the choice of Q slightly, and the proposed choice of Q is not properly motivated in my opinion. The work also does not compare with using epoch-varying losses or two-stage approaches for other methods."***
>
> A2: Yes, we agree that our proposed choice of Q is not properly motivated and the newly proposed method only differs from Balanced-Softmax in the choice of $Q$ slightly though our main purpose is to show the conclusion that classification is OT. However, However, we believe that one day, one will be able to discover a more suitable to enhance representation learning. In fact, we are currently adopting a new $Q$, which is computed using a vanilla softmax-trained "teacher" model. We present the results in the uploaded PDF in Tab. 3 and it is interesting to find a great improvement for the head classes, which inspires us to further explore the choice of $Q$.
>
> >***Q3: "The evaluations show relatively minor improvements compared to existing works, and most experiments do not report error bars so it is unclear if most improvements are significant."***
>
> A3: Thanks. We train the different models 10 times on CIFAR100 and the results are in Tab. 2 in the PDF file.
>
> >***Q4:How does (15) relate to (10)?***
>
> A4: The following are our thoughts that relate Eq.15 to Eq.10.
>
> Eq.10 provides a deblurred image by gradually optimizing Q, indicating that Q does not necessarily need to be set as a fixed matrix. Building upon the observation that a two-stage approach, where vanilla softmax is trained first and then the modified loss is trained, can yield good results, it is worth noting that directly training with the modified loss (e.g., Balanced softmax without vanilla Softmax pretraining) may lead to poorer performance. This suggests that different training stages require varying degrees of correction. Therefore, we propose Eq.15 as our loss function.
>
> >***Q5:Are (17) and (18) unique to the OT framework?***
>
> A5: To the best of our knowledge, yes, we have only encountered relevant definitions and formulas within OT. In fact, Eq.17 investigates the transformation of features between two different spaces based on coupling. It may not be immediately apparent to draw a direct connection between this problem with the traditional Bayesian perspective, which primarily focuses on studying the conditional probability $P(y|x)$.
>
> >***Q6:The work uses n for both number of iterations and dimension of polytope $\Sigma_n$ which can be confusing.***
>
> A6: Thanks for pointing it out. We will correct it in the new version.
>
> >***Q7:Proposition 3 appears to be misplaced.***
>
> A7: Prop. 3 mainly follows the dual form of Entropic OT. Please see Proposition 4.4 in [1] (Page 77 if in the same version)
>
> [1] G. Peyre and M. Cuturi. Computational optimal transport. Foundations and Trends in Machine Learning, 11(5-6):355–607, 2019.
>
> >***Q8:The notations in Figure 2 and equation (10) are quite hard to understand.***
>
> A8: Figure 2 talks about the barycenter calculated by Eq. 9 between noise and a leopard image. The first row is the results based on Entropic OT, where the barycenter images are blurred. The following two rows are RE-OT-based results setting different $Q$. We will polish the image and description. Thanks.
>
> For Eq. 10, we have interpreted in the response of A1 for the Reviewer RCtv. Due to the limited length of rebuttal, we don't repeat our answers. Thank you for your understanding.

---

> > ### Comment · Reviewer_HboX · 2023-08-17
> >
> > Thank the authors for their response. For Q1, the proposed usages of the RE-OT framework are indeed interesting and potentially fruitful. I can now agree with the authors that the proposed OT view of classification could be valuable. However, it is also the case that these newly proposed ideas have not been demonstrated in this work, so it is still unclear if they are actually useful. For Q2, it appears that the teacher model caused a quite drastic decline in accuracy in the Few case, which is somewhat unintuitive. I still do not understand the answer to Q4 and Q8, a confusion which seems to be shared by other reviewers. The rest of the answers are fine.
> >
> > Overall, I think the framework proposed in this work is of interest to NeurIPS but many parts of the work can be improved (in particular clarity and better motivation). I have read the other reviewer's comments as well as the author's rebuttal. I lean towards keeping my score unchanged, but I would look forward to see further clarifications from the authors on Figure 2 and equation (10), as well as responses from other reviewers.

---

> > > ### Author Response · Authors · 2023-08-17
> > >
> > > Thank you very much for your response. Here is our new reply:
> > >
> > > >***Q1: "However, it (OT view for classification) is also the case that these newly proposed ideas have not been demonstrated in this work, so it is still unclear if they are actually useful."***
> > >
> > > A1: We conducted a simple experiment to further demonstrate the usefulness of viewing classification as OT. In the inference phase on the testing data, we replaced the softmax (i.e. constraints within $U(\mathbf{a})$) with Sinkhorn algorithm (i.e. constraints within  $U(\mathbf{a},\mathbf{b})$), where $\mathbf{b}$ represents the assumed ratio in the testing data (e.g., long-tailed, uniform, or reverse long-tailed distribution). The accuracy (\%) results are shown below:
> > >
> > > | Models:valilla softmax CE  Loss | LT   | Uniform | Reverse LT |
> > > | ------------------------------- | ---- | ------- | ---------- |
> > > | Inference with Softmax          | 58.7 | 40.8    | 23.5       |       |
> > > | Inference with Sinkhorn         | 59.1 | 46.6    | 39.6    |
> > >
> > > | Models:our RE-OT based Loss | LT  | Uniform | Reverse LT |
> > > | --------------------------- | --- | ------- | ---------- |
> > > | Inference with Softmax          | 60.4 | 47.4   | 35.1      |       |
> > > | Inference with Sinkhorn         | 60.4| 47.9    | 40.9    |
> > >
> > > In the above experiments, we reconfigured the testing data to have long-tailed (IF=10), uniform (unchanged), and reverse long-tailed (IF=10) distributions on CIFAR100. We found that using Sinkhorn as the inference method during the testing process can lead to significant improvements, particularly for the model trained with vanilla softmax. These experiments validate the effectiveness of treating classification as Optimal Transport during the testing inference stage.
> > >
> > > >***Q2: Clarifications on Figure 2 and equation (10) .***
> > >
> > > A2: Thanks. We can provide further clarifications in the form of a logical chain.
> > >
> > > Assumption： Image $a$,  Image $b$  and Barycenter $c$
> > >
> > > 1) $P^\epsilon$: the transport $a\to b$ and $(P^\epsilon)^\top$: the transport $b\to a$
> > >
> > > 2) $P_1$: the transport $a\to c$ and $P_2$: the transport $b\to c$
> > >
> > > 3) the regularization $H_Q(P)$  push the solution $P$ to $Q$
> > >
> > > 4) If we set $Q=P^\epsilon$, $P_1$ will be more close to $P^\epsilon$, making $c$ more close to $b$.
> > > (The reason is $(P^\epsilon)^\top 1_n =b$, $P_1$ is close to $P^\epsilon$, then $P_1^\top 1_n$ is close to $b$)
> > >
> > > 5) If we set $Q=(P^\epsilon)^\top$, $P_2$ will be more close to $(P^\epsilon)^\top$, making $c$ more close to $a$.
> > >
> > > 6) We set $Q= \lambda P^\epsilon+(1-\lambda)(P^\epsilon)^\top$, $c$ will be more close to $a$ and $b$. Specially, when $\lambda\to 0$, then $Q=(P^\epsilon)^\top$. We can get $c$ more close to $a$ from (4), while $\lambda\to 1$, then $Q=P^\epsilon$, we can get $c$ more close to $b$ from (5).

---

### Official Review · Reviewer_46Rh · 2023-07-02

**Soundness:** 3 good
**Presentation:** 3 good
**Contribution:** 2 fair
**Rating:** 6
**Confidence:** 3

**Summary:**

The authors propose Relative Entropic Optimal Transport, which allows to incorporate prior information matrix to the learning of optimal transport plan. After studying its theoritical properties, they adapt REOT to the long-tailed classification problem and establish the connection between optimal transport and classification. Finally, they illustrate the effectiveness of the proposed method via experiments on long-tailed classification and representation learning.

**Strengths:**

The authors have well established the connection between REOT and various classification loss functions. They also clearly demonstrate how REOT can be used in long-tailed problem. The proposed method also shows strong performance in many long-tailed experiments.

**Weaknesses:**

I would say the contribution to the OT theory is quite limited and incremental because all theoritical results of REOT are straightforward adaptation from the EOT. The Prop 1 and its proof need serious revision because there are various typos, which leave me doubt on the correctness of the theoritical result and proof.

- In Eq 20, 21, there is no consistent use of f and f', g and g'.

- In Eq 21, there should be no epsilon. This makes the form of Qtilda incorrect because it depends on epsilon.

- In Eq 22, 23, there is no epsilon.

- In Eq 23, 24, there is no consistent use of f and f', g and g'.

- Typos in Eq 25.

- In Eq 25, I fail to see the last equality and to see why we can set f and g in such ways to get the equality. I feel like this is a circular reasoning.

- In Prop 1, the authors state that when Qtilda = a otimes b, then the solution of REOT and EOT coincides (which is not true in general), while in the proof, they consider Qtilda = one matrix, which makes the two solution coincide.

- There are also typos in Eq 42, 43.

Despite some concerns on the theoritical result, I feel that it can be fixed without impacting the effectiveness and soundness of the proposed method.

**Questions:**

- I don't really understand the context of Figure 4: can the authors provide more description of the dataset and experiment?

- Small remark: the authors might also want to add relevant reference to the barycentric projection:
1) Ferradans, S.; Papadakis, N.; Peyré, G.; and Aujol, J.-F. 2014. Regularized Discrete Optimal Transport. SIAM Journal on Imaging Sciences, 7(3): 1853–1882.

2) Courty, N.; Flamary, R.; Tuia, D.; and Rakotomamonjy, A. 2016. Optimal transport for domain adaptation. IEEE Transactions on Pattern Analysis and Machine Intelligence, 39: 1853–1865.



**Limitations:**

The authors do discuss the limitations of their work.

---

> ### Author Rebuttal · Authors · 2023-08-09
>
> Thank you for spending your valuable time on this paper. We hope that our responses can address the concerns you may have.
>
> >***Q1: "In Prop 1, the authors state that when Qtilda = a otimes b, then the solution of REOT and EOT coincides (which is not true in general), while in the proof, they consider Qtilda = one matrix, which makes the two solution coincide."***
>
> A1: Thank you for the comments. We apologize for the misunderstanding. We have corrected the errors in the proof of Prop. 1, and you can see the updated version in the uploaded PDF if you are interested.  Here are our responses to your points:
>
> 1. $\tilde{Q}$ belongs to $U(a,b)$ due to the optimization problem $\tilde{Q}=\arg\min_P -H_Q(P), s.t. P\in U(a,b)$.
>
> $\quad$ So, one can set $Q=\mathbf{1}_{n\times m}$ or ${Q}=a\otimes b$.
>
> $\quad$  But it is wrong to set $\tilde{Q}=\mathbf{1}_{n\times m}$.
>
> $\quad$  Specifically, when $Q=\mathbf{1}_{n\times m}$, we can simply obtain its corresponding $\tilde{Q}=a\otimes b$.
>
> 2. Most papers adopt $Q=\mathbf{1}_{n\times m}$ for the entropic formulation. However, a few papers do use the form ${Q}=a\otimes b$. Please see [1]. These two formulations are exactly equivalent, and the purpose of Prop.1 is to demonstrate that different $Q$ can have equal RE-OT solutions when they have the same $\tilde{Q}$.
>
> $\quad$  We have provided simple numerical experiments **in Fig. 2** in the uploaded PDF to show the **equivalence of $P_{Q}$ and $P_{\tilde{Q}}$** .
>
> 3. Sorry for the typos in the proof that may cause misunderstandings. We make corrections to the proof of Prop. 1 **in the uploaded PDF (Fig. 1)**.
>
> [1] Statistical bounds for entropic optimal transport: sample complexity and the central limit theorem, NeurIPS 2019.
>
> >***Q2: "I don't really understand the context of Figure 4: can the authors provide more description of the dataset and experiment?" ***
>
> A2: Thank you. The following is our description of the dataset and experiment of Fig.4. Exactly, in the experiment of Fig.4, we first train the models using vanilla softmax, focal loss, and our loss on the Cifar10-LT dataset. We consider the logits as the sample features (in theory, we could also choose intermediate features of the network instead) and save the logits from the three head and three tailed classes, along with the corresponding predicted couplings, as matrices.
>
> Then, we calculate the barycenters of the labels using Eq.18, which effectively computes the weighted average of the features. We concat the logits and barycenters as a new matrix and use it to calculate t-SNE results for the training data, as shown in the first row of Fig.4. For the testing data, we concat the testing logits with the barycenters calculated from the training data and use t-SNE dimensional reduction, as shown in the second row in Fig.4. By comparing the first row and the second row of Fig.4, we can observe the differences in the positions of the barycenters, which are actually caused by shift in the feature distributions.
>
> >***Q3: About typos and remark about adding relevant reference to the barycentric projection***
>
> A3：Thanks again. We will correct our mistakes and add barycentric projection references.

---

> > ### Comment · Reviewer_46Rh · 2023-08-18
> > **Response to the authors**
> >
> > I thank the authors for their reponse.
> > I will leave more comments in the general discussion.

---

### Official Review · Reviewer_pKsm · 2023-07-06

**Soundness:** 2 fair
**Presentation:** 3 good
**Contribution:** 3 good
**Rating:** 5
**Confidence:** 3

**Summary:**

This paper addresses the problem of unbalanced classification using a variant of optimal transport called Relative Entropic Optimal Transport (RE-OT), which alter the Sinkhorn distance with the prior information into the coupling solution. Experimental results across different domains validate the efficacy of their approach.

**Strengths:**

The paper is easy to follow. The idea of using Barycentric projection used in previous work in domain adaptation with OT has been adapted  the context of (unbalanced) classification.

**Weaknesses:**

The proposed method much depends on the prior distribution Q which is supposed as the label distributions of the data. However, specifying correct Q is impossible in the reality. The author should show the effects of wrongly specified Q by experimental results. Otherwise, the proposed method has little applicable in the real world application.

**Questions:**

A recent work [1] has introduced a principled framework for using optimal transport for class-Imbalanced classification problem. Could you please differentiate the current work with this work in terms of theorical and experimental (if time allows) aspects.
[1] Jin, Lianbao, Dayu Lang, and Na Lei. "An Optimal Transport View of Class-Imbalanced Visual Recognition." International Journal of Computer Vision (2023): 1-19.


**Limitations:**

The paper already mentioned its limitation: "It assumes known prior label distribution testing data, which is often unknown in real-world scenarios"

---

> ### Author Rebuttal · Authors · 2023-08-09
>
> Thank you very much for taking the time to review this paper. Below are our responses.
>
> >***Q1: Difference to [1]***
>
> A1: [1] views neural networks as mappings in optimal transport and proposes that these mappings need to adapt due to the difference between the training and testing label distributions. In terms of technical details, it primarily investigates the dual form of Kantorovich OT and establishes a connection between the gradient of the Kantorovich functional and cross-entropy. Modifying the Kantorovich functional for the inconsistency between the training and testing label distributions can improve testing predictions. In contrast, our paper shows the following differences:
>
> 1. We do not start from the dual form but directly focus on the primary OT formulation, leading to the conclusion that Softmax is a special case.
> 2. The theoretical approximation proposed in [1] relates gradients between cross-entropy and the Kantorovich functional, whereas our equivalence lies between Softmax and Kantorovich coupling, which is the main difference.
> 3. Our paper is not limited to long-tailed problems alone. The main purpose of Section 4 is to illustrate that modified Softmax-CrossEntropy or other classification losses, from the OT  perspective, can be viewed as different OT settings. Consequently, we can conclude that classification can be understood as OT. In future works, we can then incorporate other theoretical advancements of OT into classification tasks. We provide some examples of using the OT perspective to improve classification in the response to Reviewer Cbtk in A3.
>
> >***Q2: About the $Q$ in real world.***
>
> A2: Yes, RE-OT is dependent on the prior $Q$ which may not exist in real-world. Here we give our ideas that may address your concerns:
>
> 1. The optimal $Q$ is indeed unknown, and the known label ratios serve as a surrogate choice for (long-tailed) classification. However, we believe that one day, one will be able to discover a more suitable $Q$ to enhance representation learning. In fact, we are currently adopting a new $Q$, which is computed using a vanilla softmax-trained "teacher" model. We present the results in the uploaded PDF in Tab. 3 and it is interesting to find a great improvement for the head classes, which inspires us to further explore the choice of $Q$.
>
> 2. The purpose of Sec.4 is mainly to give the perspective that the classification is OT. Many variants of OT can be introduced to the classification problem with this OT perspective (e.g. optimal partial transport with the constrain $\\{\mathbf{P}\mathbf{1}\leq \mathbf{a}, \mathbf{1}^\top \mathbf{P}\mathbf{1}=s\\}$. See more examples in A3 of the rebuttal for reviewer RCtv.

---

> ### Author Response · Authors · 2023-08-21
>
> Dear Reviewer pKsm:
>
>  As the discussion deadline approaches, we would greatly appreciate it if you could provide feedback on our response regarding addressing the concerns. We are also open to any further questions or suggestions that you may have.
>
>  Thank you once again for your time and attention.

---

### Official Review · Reviewer_RCtv · 2023-07-07

**Soundness:** 3 good
**Presentation:** 3 good
**Contribution:** 3 good
**Rating:** 5
**Confidence:** 4

**Summary:**

This works proposed a new view to the classification problem through the lens of Optimal Transpot. They propose a new variant of OT, says Relative-OT, by changing the KL regularizer constraints in a given distribution. By some special properties of KL divergence, the works show a relationship betwen the entropic relative OT and the entropic OT in Proposition 1 and 2.

In the application of finding barycenter of multiple distributions, the authors proposed to iteratively change the relative distribution in the regularization to obtain a "smooth" (maybe meaningful) barycenter.

Another application is the long-tailed recognition problem. They define a cost matrix based on the logit value of the prediction neural network. Then they view the classification problem like a matching problem between data and labels. They aim to find the transport plan satisfying both constraints: one is from the true label, and one is from  the "transport cost" between data and probability value of a NN. For a long-tail issue, a relative distribution is set adaptively in the training process.

Section 5 shows the experiment results on LongTail data set and unbalanced data sets, like CIFAR10-LT, CIFAR100-, ImageNet_LT, molecule data etc.

**Strengths:**

I believe that the idea of using OT in the classification problem as presented is new.

Experiment results show an improvement in performance compared to other methods but not in all cases.

**Weaknesses:**

I myself still do not get the justification for using the OT in classification problem. I mean all the use of relative distribution, iteration, restate the classification as two problems of finding optimal plan and distribution etc. I see some toy example of blur images for explanation, but apart from that I do not find any others, maybe I miss some parts of the paper. Here I mean the theory and intuition behind it, rather than just assemble OT and classification problem in the presented way. When more parameters involves in the training process of the model, it is more likely that we will obtain better performance.

**Questions:**

1. I do not understand the claim in equation (10). Could the author elaborate it?

2. In Table 3, case of Many, the performance of proposed method is worse than some other methods. Could the authors provide explanations?



Minors: typos Qij line 164, proposition 1: "and are"

**Limitations:**

It is fine.

---

> ### Author Rebuttal · Authors · 2023-08-09
>
> Thank you for your valuable time in reviewing this paper. Here are our responses.
>
> >***Q1: "I do not understand the claim in equation (10). Could the author elaborate it?""***
>
> A1: Certainly. Here is our explanation. Assume that $P^\epsilon$ is a coupling from image $a$ to image $b$, and we can obtain the normalized pixel values $b_j=\sum_i P^{\epsilon}_{ij}$.
>
> When $b_j$ is small (indicating a white region at position $j$), the corresponding entries $P^{\epsilon}_{:,j} $  will also be small.
>
> Consequently, when computing the barycenter $c$, by setting $Q=P^{\epsilon}$ as the coupling from $a$ to $c$, the solution $P_{:,j}$ will be influenced by $Q$, resulting in a small value for $c_j$. This leads to a non-blurry image. Considering the transportation from $b$ to $a$, we interpolate between $P^{\epsilon}$ and its transpose to obtain the final coupling matrix $Q$.
>
> >***Q2: In Table 3, case of Many, the performance of proposed method is worse than some other methods. Could the authors provide explanations?***
>
> A2: This scenario is quite common also for many peer methods in long-tailed experiments. Essentially, it represents a tradeoff between the head and tailed samples. To achieve higher accuracy for the tailed classes, there is a certain sacrifice in accuracy for the head class. However, the accuracy of the whole data tends to improve as a result.
>
> >***Q3: "I myself still do not get the justification for using the OT in classification problem."***
>
> A3: We think that viewing classification as OT is highly valuable. Although we mentioned in the future work section that the OT perspective might help approach open-set problems, we can elaborate more on this direction if you are willing to hear about it. Here are a few directions we can do from the OT perspective yet less possible in the traditional Bayesian way:
>
> **1.  Change in the coupling constraint  $\\{ P: P1=a \\}$**.  We can adopt other constraints instead of $\\{ P: P1=a \\}$, such as using the constraints in (modified) Optimal Partial Transport (OPT) with $\\{ P: P1 \leq a,1^\top P1=s \\}$. This allows the model that is not just choosing one from all classes and enables rejection of classification if the model determines that a sample cannot be classified into any of the candidate labels.
>
> **2. Change in label selection in one model**. In the OT perspective, classification is not understood in terms of $P(y|x)$ but rather from a matching perspective. Therefore, when we have representations of samples and labels, we can freely choose different label sets to form different classification tasks. For example, during training, we classify into ${y_1,y_2,\dots,y_{10}}$, but during testing, we classify into ${y_3,y_7,y_{10}}$. Additionally, if the label representation is based on NLP embedding models, we can include previously unseen labels in the classification testing phase, potentially addressing zero-shot learning problems without the need for additional parameter training.
>
> **3. Generalization of Softmax using other regularization.** The current softmax formulation is essentially based on the Entropic Regularization of OT. However, OT regularization goes beyond just the entropic regularization and includes other regularizations such as L2, Tsallis entropies, or divergence-based ones. This opens up possibilities for generalizing softmax.
>
> **4. Classification cross-representation models using Gromov-Wasserstein Distance.** From the OT perspective, classification between the features of samples and labels, based on the Gromov-Wasserstein Distance, can be performed in different feature spaces. For example, the sample features may be in 100 dimensions while the label features are in 20 dimensions. This can be helpful for preserving privacy as Gromov-Wasserstein Distance only requires similarity between samples and labels.
>
> Due to the various variants and theories within OT, considering classification as OT naturally allows us to incorporate existing OT knowledge into the field of classification, which is the ongoing work we are conducting. Only through this OT perspective, it becomes possible in the future to develop models that can "Classify Anything," including different task selections, variable classification constraints, and more in one model. Therefore, the main purpose of this paper is to provide a different perspective on classification models and offer conceptual assistance to general recognition models.
>
> >***Q4: About Typos.***
>
> A4: Thank you for your feedback. We will make corrections in the final version.

---

> > ### Comment · Reviewer_RCtv · 2023-08-20
> > **reply to the authors**
> >
> > I would like to thank the authors for their answers. Because I am not satisfied with the current presentation form of the paper, I will keep my score unchanged.

---

### Official Review · Reviewer_Cbtk · 2023-08-02

**Soundness:** 3 good
**Presentation:** 2 fair
**Contribution:** 3 good
**Rating:** 6
**Confidence:** 4

**Summary:**

The paper introduces the relative entropic (RE) regularization for optimal transport (OT) problems. Instead of the traditional entropy regularizer (Cuturi'13), the RE-OT is defined through a given prior matrix $\boldsymbol{Q}$, which guides the matching between source and target distributions. Using the approach of Inverse RE-OT, the authors derive a similar form to softmax-based cross-entropy loss to train long-tailed data in computer vision pipelines.

**Strengths:**

- The paper introduces RE-OT, a new variant of entropic regularization for OT.
- The RE-OT enjoys some theoretical properties including a static Schrodinger form and a dual formulation as for the standard entropic regularizer.
- The paper investigates the Inverse RE-OT for long-tailed recognition tasks and shows that these tasks can be formulated as matching perspectives with OT.
- Extensive numerical experiments on image classification and molecule classification.

**Weaknesses:**

- The RE-OT is strongly dependent on the prior $\boldsymbol{Q}$, that can be an additive hyperparameter.
- The Inverse RE-OT approach is limited since one has to have a good knowledge of the supervision $\tilde{\boldsymbol{P}}.$

**Questions:**

### Questions:
- Does the prior guide $\boldsymbol{Q}$ belong, in general, to the polytope $U(a,b)$?
- In the experiments, how the parameter $\epsilon$ was chosen?
- During training, the constant $c$ is it fixed? How can guarantee the positiveness of $\boldsymbol{C}_{ij}$?
- In L554, the last approximation is obtained by a Taylor expansion?

### Typos
Below I list some typos to be corrected:
- L38: involves learning the Inverse OT: add a reference
- L105: $\tilde{\boldsymbol{P}}$ is not defined
- L163:  bold $Q$
- L164: $\tilde{\boldsymbol{Q}}_{ij}$
- L165: ${\boldsymbol{P}}^\epsilon_{\tilde{\boldsymbol{Q}}}$
- L173, L175: bold $K$
- L180: in Eq (10), there is no transpose
- L184: as $\lambda$ changes.
- L192 - L202: avoid to use $n$ as  the iterated index ($^{(n)}$)
- L210: constant $c$ is not commented
- L211: in Eq (14), the cost matrix $\boldsymbol{C}$ depends in $\theta$
- L222: is an epoch
- L240: in Eq (16) there is a dependency on $\theta$. Also, a minus sign and a factor $n$ are missing, i.e.
\begin{equation}
\min_{\theta} L = - \sum_{i,j} \tilde{\boldsymbol{P}}_{ij} \log \frac{\cdots} {n \cdots}
\end{equation}
- L258: deep graph matching: add a reference
- L282, L283: croponding --> corresponding
- L503: Eqs (22) and (23): missing dependency on $\epsilon$
- L507: multiplier --> multipliers
- L507: $\tilde{\boldsymbol{Q}}_{ij}$
- L508: in Eq (25), $\boldsymbol{P}^\epsilon_Q = \cdots {\boldsymbol{Q}} diag(e^{g'/\epsilon})$
- L510: \boldsymbol{Q}
- L522: $-\boldsymbol{C}_{ij}$
- L542 and L543: depedency on $\theta$
- L547: Eq (44)
$$
\min_{\theta} L = - \sum_{i,j} \tilde{\boldsymbol{P}}_{ij} \log \frac{\cdots} {n \cdots}
$$

---

> ### Author Rebuttal · Authors · 2023-08-09
>
> We appreciate your valuable time and comments and we hope the following answers can address your concerns.
>
> >***Q1: Does the prior guide Q belong, in general, to the polytope U(a,b)***
>
> A1: No. For example, $Q$ can be set as $1_{n×m}\notin U(a,b)$, and then the RE-OT problem degenerates into Entropic OT. However, the resulting $\tilde{Q}$ computed based on $Q$ belongs to the polytope $U(a,b)$. As demonstrated in Prop.1 (2), the use of $Q$ and $\tilde{Q}$ is essentially equivalent.
>
> >***Q2: In the experiments, how the parameter $\epsilon$ was chosen?***
>
> A2：We simply set $\epsilon=1$. Here $\epsilon$ is exactly equivalent to the temperature $\tau$ in the Softmax function. In our paper, we did not modify the temperature and followed the setting of Vanilla Softmax, where we simply set $\epsilon=1$.
>
> >***Q3: During training, the constant c is it fixed? How can guarantee the positiveness of $C_{ij}$***
>
> A3:  $c$ is fixed but is independent of training.  Here, $c$ refers to a sufficiently large number that exists only in theory. We can easily observe that $C_{ij}=c-l_{ij}>0$ when $c$ is sufficiently large, ensuring the positivity of $C_{ij}$. However, in practical computations, when performing row normalization, we can obtain $e^{-(c-l_{ij})}/\sum_k e^{-(c-l_{ik})}=e^{l_{ij}}/\sum_k e^{l_{ik}}$. The results are independent of $c$.
>
> >***Q4: In L554, the last approximation is obtained by a Taylor expansion?***
>
> A4: Yes, the last approximation is obtained through Taylor expansion as $\epsilon$ approaches 0 from the positive side. This derivation is inspired by Eq. 6 in [1]. We will clarify it in the new version.
>
> >***Q5: The Inverse RE-OT approach is limited since one has to have a good knowledge of the supervision $\tilde{P}$***
>
> A5: Yes, one has to know $\tilde{P}$ for Inverse RE-OT; otherwise, we cannot optimize Inverse RE-OT for representation learning. However, for a classification task, $\tilde{P}$ is assumed to be known as the labels of samples.
>
> >***Q6:The RE-OT is strongly dependent on the prior Q, that can be an additive hyperparameter.***
>
> A6: Yes, RE-OT is dependent on the prior $Q$ which can be an additive hyperparameter. However, we try to address your concern about the application of OT and (long-tailed) classification fields:
>
> (1) For OT, $Q$ can be **computed** based on practical needs, instead of artificially adding a hyperparameter. For instance, as shown in Eq. 10, by setting $Q$ in a specific way, we can deblur the barycenter of images, which is not achievable with traditional Entropic OT.
>
> (2) For (long-tailed) classification, the optimal $Q$ is indeed unknown, and the known label ratios serve as a surrogate choice. However, we believe that one day, we will be able to discover a more suitable $Q$ to enhance representation learning. In fact, we are currently adopting a new $Q$, which is computed using a vanilla softmax-trained "teacher" model. We present the results in the uploaded PDF in Tab. 3 and it is interesting to find a great improvement for the head classes, which inspires us to further explore the choice of $Q$.
>
> >***Q7: About Typos.***
>
> A7: Thank you for your feedback. We will make corrections in the final version.
>
> [1]  F. Wang and H. Liu. Understanding the behavior of contrastive loss. In Proceedings of the IEEE/CVF Conference on Computer Vision and Pattern Recognition, pages 2495–2504, 2021

---

> > ### Comment · Reviewer_Cbtk · 2023-08-15
> > **Thank you for the response**
> >
> > I thank the authors for their efforts in the rebuttal.

---

### Author Rebuttal · Authors · 2023-08-09

We sincerely appreciate the reviewers for investing their valuable time and providing insightful comments on our paper. Overall, the reviewers found our work to be novel (Cbtk, RCtv), with promising experimental results (Cbtk, RCtv, HboX), and easy to follow (pKsm). Additionally, they acknowledged the theoretical properties (Cbtk), the meaningful barycenter calculation (RCtv), and the connections between REOT and various classification losses (46Rh, HboX). We are grateful for the recognition and encouragement from the reviewers, and their comments have inspired us to further improve our work.

Simultaneously, the reviewers have provided several critical insights and improvement suggestions. In response to these, we have carefully considered their feedback and provided our response. For instance, one common concern raised by multiple reviewers is the practical significance of viewing classification as OT. We have addressed this concern by offering explanations and providing several examples to illustrate the interesting and meaningful aspects of this perspective. We firmly believe that our work will bring further inspiration to the readers.

---

### Author Response · Authors · 2023-08-17

Dear AC and Reviewers,

  Thank you for your valuable efforts and the constructive suggestions provided by the reviewers. As the discussion deadline approaches, we would like to request your feedback on our response to address the reviewers' concerns. We are also open to any further questions or suggestions that you may have.

 Thank you once again for your time and attention.

---

### Author Response · Authors · 2023-08-21
**Summary of author-reviewer discussion**

Dear AC:
Thanks for your effort in organizing the review of our submission. Here is our summary of the author-reviewer discussion:

Reviewer **Cbtk** acknowledges our proposed RE-OT as a new variant of OT and appreciates the theoretical properties presented in this paper. The reviewer agrees with the OT perspective for long-tailed classification and finds our experiments extensive. He/She votes for a **Weak Accept (6)**. Their concern is the strong dependence on $Q$ and $\tilde{P}$ in our RE-OT approach. In response, we clarify that $\tilde{P}$ is known for supervised tasks, and the choice of $Q$ can be flexible to improve representation learning, e.g. adopting a teacher model to calculate $Q$.

Reviewer **RCtv** finds the idea of using OT in classification to be novel, and the experimental results demonstrate improvements over the baselines. He/She votes for a **Borderline Accept (5)** but express a need for further justification for using OT in classification. In our response, we highlight the advantages of viewing classification as OT and provide examples where classification via OT outperforms traditional Bayesian approaches. However, reviewer RCtv maintains their score unchanged due to concerns about the current paper's presentation. We assure him/her that we will make every effort to improve the writing in the final version here.

Reviewer **pKsm** finds our paper easy to follow, and the barycentric projection is adaptive to (unbalanced) classification. He/She votes for a **Borderline Accept (5)** and the main concern is the dependence on $Q$ in reality. We provide detailed responses to address the raised concerns but have not received any feedback. We sincerely hope that Reviewer pKsm could adjust the rating if the concerns and questions are addressed.

Reviewer **46Rh** agrees with the connection between RE-OT and various classification loss functions but mainly concerns about the theoretical results of Proposition 1. Thus, he/she initially gave a **Borderline Accept (5)** rating. We provide a detailed response, and after reading it, the reviewer raises the rating to **Weak Accept (6)**.

Reviewer **HboX** finds our RE-OT theory novel, clear, and easily understandable with visualizations and explanations. The reviewer also agrees with the connection between OT and classification as summarized in Table 1. He/She gives a **Borderline Accept (5)** rating due to some concerns and questions (e.g., why RE-OT provides a useful way for tackling classifications). After the discussions, although the rating remains unchanged, the reviewer thinks the framework proposed in this work is of interest to NeurIPS, but many parts of the work can be improved. The reviewer also agrees that the proposed OT view of classification is indeed interesting, potentially fruitful, and valuable. We truly appreciate the reviewer's time and effort spent on this paper.

At last, we sincerely thank the reviewer and AC for investing their valuable time and efforts in this paper.

---

### Decision · Program_Chairs · 2023-09-21

**Decision:**

Accept (poster)

**Comment:**

The reviewers found the paper interesting, with new entropic OT and inverse OT formulation and with surprising relations with (unbalanced) classification but had a few concerns such as practicality of the method, lack of details for numerical experiment, and writing, which lead to a borderline positive score.

The authors did a detailed rebuttal and provided additional experiments. The practicality concerns were discussed by the authors and they provided a new experiment with a teacher model. The missing references have been discussed well. The rebuttal was appreciated by the reviewers (one increased their score).

The final decision is to accept the paper but to strongly suggest that the authors take into account the comments from the reviewers, integrate to the paper the new results and discussion and do a major proof checking of the paper since that need was noted by most reviewers.